# Task-Adaptive Neural Network Search with Meta-Contrastive Learning

**Wonyong Jeong**[1,2*†] **Hayeon Lee**[1,2*†] **Geon Park**[1,2*†]
**Eunyoung Hyung**[1,2†] **Jinheon Baek**[1] **Sung Ju Hwang**[1,2]
KAIST[1], AITRICS[2], Seoul, South Korea
wyjeong@kaist.ac.kr, hayeon926@kaist.ac.kr, geon.park@kaist.ac.kr
ey0301.hyung@samsung.com, jinheon.baek@kaist.ac.kr, sjhwang82@kaist.ac.kr

## Abstract

Most conventional Neural Architecture Search (NAS) approaches are limited in that they only generate architectures without searching for the optimal parameters. While some NAS methods handle this issue by utilizing a supernet trained on a large-scale dataset such as ImageNet, they may be suboptimal if the target tasks are highly dissimilar from the dataset the supernet is trained on. To address such limitations, we introduce a novel problem of *Neural Network Search* (NNS), whose goal is to search for the optimal pretrained network for a novel dataset and constraints (e.g. number of parameters), from a model zoo. Then, we propose a novel framework to tackle the problem, namely *Task-Adaptive Neural Network Search* (TANS). Given a model-zoo that consists of network pretrained on diverse datasets, we use a novel amortized meta-learning framework to learn a cross-modal latent space with contrastive loss, to maximize the similarity between a dataset and a high-performing network on it, and minimize the similarity between irrelevant dataset-network pairs. We validate the effectiveness and efficiency of our method on ten real-world datasets, against existing NAS/AutoML baselines. The results show that our method instantly retrieves networks that outperform models obtained with the baselines with significantly fewer training steps to reach the target performance, thus minimizing the total cost of obtaining a task-optimal network. Our code and the model-zoo are available at https://github.com/wyjeong/TANS.

## 1 Introduction

*Neural Architecture Search* (NAS) aims to automate the design process of network architectures by searching for high-performing architectures with RL [76, 77], evolutionary algorithms [43, 11], parameter sharing [6, 42], or surrogate schemes [38], to overcome the excessive cost of trial-and-error approaches with the manual design of neural architectures [47, 23, 27]. Despite their success, existing NAS methods suffer from several limitations, which hinder their applicability to practical scenarios. First of all, the search for the optimal architectures usually requires a large amount of computation, which can take multiple GPU hours or even days to finish. This excessive computation cost makes it difficult to efficiently obtain an optimal architecture for a novel dataset. Secondly, most NAS approaches only search for optimal architectures, without the consideration of their parameter values. Thus, they require extra computations and time for training on the new task, in addition to the architecture search cost, which is already excessively high.

For this reason, supernet-based methods [8, 37] that search for a sub-network (subnet) from a network pretrained on large-scale data, are attracting more popularity as it eliminates the need for additional

---

[*]Equal contribution.
[†]This work was done while the author was interning at AITRICS.

35th Conference on Neural Information Processing Systems (NeurIPS 2021).

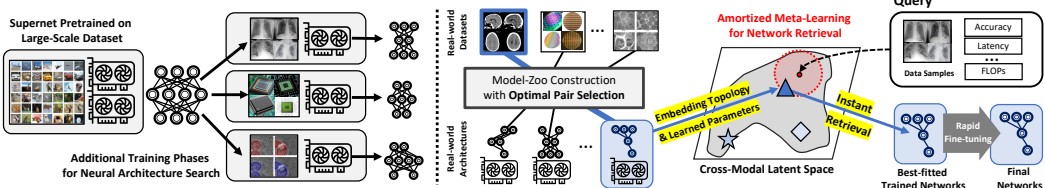

Figure 1: **Comparison between conventional NAS and our method**: Conventional supernet-based NAS approaches (Left) sample subnets from a fixed supernet trained on a single dataset. TANS (Right) can dynamically select the best-fitted neural networks that are trained on diverse datasets, adaptively for each query dataset.

training. However, this approach may be suboptimal when we want to find the subnet for a dataset that is largely different from the source dataset the supernet is trained on (e.g. medical images or defect detection for semiconductors). This is a common limitation of existing NAS approaches, although the problem did not receive much attention due to the consideration of only a few datasets in the NAS benchmarks (See Figure 1, left). However, in real-world scenarios, NAS approaches should search over diverse datasets with heterogeneous distributions, and thus it is important to *task-adaptively* search for the architecture and parameter values for a given dataset. Recently, MetaD2A [31] has utilized meta-learning to learn common knowledge for NAS across tasks, to rapidly adapt to unseen tasks. However it does not consider parameters for the searched architecture, and thus still requires additional training on unseen datasets.

Given such limitations of NAS and meta-NAS methods, we introduce a novel problem of *Neural Network Search* (NNS), whose goal is to search for the optimal pretrained networks for a given dataset and conditions (e.g. number of parameters). To tackle the problem, we propose a novel and extremely efficient task-adaptive neural network retrieval framework that searches for the optimal neural network with both the architecture and the parameters for a given task, based on cross-modal retrieval. In other words, instead of searching for an optimal architecture from scratch or taking a sub-network from a single super-net, we *retrieve* the most optimal network for a given dataset in a task-adaptive manner (See Figure 1, right), by searching through the model zoo that contains neural networks pretrained on diverse datasets. We first start with the construction of the model zoo, by pretraining state-of-the-art architectures on diverse real-world datasets.

Then, we train our retrieval model via amortized meta-learning of a cross-modal latent space with a contrastive learning objective. Specifically, we encode each dataset with a set encoder and obtain functional and topological embeddings of a network, such that a dataset is embedded closer to the network that performs well on it while minimizing the similarity between irrelevant dataset-network pairs. The learning process is further guided by a performance predictor, which predicts the model's performance on a given dataset.

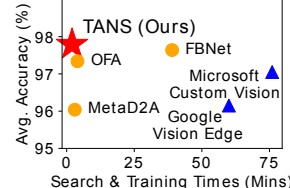

Figure 2: Comparison with NAS (orange) & AutoML (blue) baselines on 5 Real-world Datasets

The proposed *Task-Adaptive Network Search* (TANS) largely outperforms conventional NAS/AutoML methods (See Figure 2), while significantly reducing the search time. This is because the retrieval of a trained network can be done instantly without any additional architecture search cost, and retrieving a task-relevant network will further reduce the fine-tuning cost. To evaluate the proposed TANS, we first demonstrate the sample-efficiency of our model zoo construction method, over construction with random sampling of dataset-network pairs. Then, we show that the TANS can adaptively retrieve the best-fitted models for an unseen dataset. Finally, we show that our method significantly outperforms baseline NAS/AutoML methods on **real-world datasets** (Figure 2), with incomparably smaller computational cost to reach the target performance. In sum, our main contributions are as follows:

- We consider a novel problem of *Neural Network Search*, whose goal is to search for the optimal network for a given task, including both the architecture and the parameters.
- We propose a novel cross-modal retrieval framework to retrieve a pretrained network from the model zoo for a given task via amortized meta-learning with constrastive objective.
- We propose an efficient model-zoo construction method to construct an effective database of dataset-architecture pairs, considering both the model performance and task diversity.
- We train and validate TANS on a newly collected large-scale database, on which our method outperforms all NAS/AutoML baselines with almost no architecture search cost and significantly fewer fine-tuning steps.

## 2 Related Work

**Neural Architecture Search**  *Neural Architecture Search* (NAS), which aims to automate the design of neural architectures, is an active topic of research nowadays. Earlier NAS methods use non-differentiable search techniques based on RL [76, 77] or evolutionary algorithms [43, 11]. However, their excessive computational requirements [44] in the search process limits their practical applicability to resource-limited settings. To tackle this challenge, one-shot methods share the parameters [42, 6, 35, 65] among architectures, which reduces the search cost by orders of magnitude. The surrogate scheme predicts the performance of architectures without directly training them [38, 75, 54], which also cuts down the search cost. Latent space-based NAS methods [38, 54, 67] learn latent embeddings of the architectures to reconstruct them for a specific task. Recently, supernet-based approaches, such as OFA [8], receive the most attention due to their high-performances. OFA generates the subnet with its parameters by splitting the trained supernet. While this eliminates the need for costly re-training of each searched architecture from scratch, but, it only trains a fixed-supernet on a single dataset (ImageNet-1K), which limits their effectiveness on diverse tasks that are largely different from the training set. Whereas our TANS task-adaptively retrieves a trained neural network from a database of networks with varying architectures trained on diverse datasets.

**Meta-Learning**  The goal of meta-learning [55] is to learn a model to generalize over the distribution of tasks, instead of instances from a single task, such that a meta-learner trained across multiple tasks can rapidly adapt to a novel task. While most meta-learning methods consider few-shot classification with a fixed architecture [56, 20, 48, 40, 33, 30], there are a few recent studies that couple NAS with meta-learning [46, 34, 17] to search for the well-fitted architecture for the given task. However, these NAS approaches are limited to small-scale tasks due to the cost of roll-out gradient steps. To tackle this issue, MetaD2A [31] proposes to generate task-dependent architectures with amortized meta-learning, but does not consider parameters for the searched architecture, and thus requires additional cost of training it on unseen datasets. To overcome these limitations, our method retrieves the best-fitted architecture with its parameters for the target task, by learning a cross-modal latent space for dataset-network pairs with amortized meta-learning.

**Neural Retrieval**  Neural retrieval aims to search for and return the best-fitted item for the given query, by learning to embed items in a latent space with a neural network. Such approaches can be broadly classified into models for image retrieval [21, 14, 66] and text retrieval [73, 9, 63]. Cross-modal retrieval approaches [32, 74, 58] handle retrieval across different modalities of data (e.g. image and text), by learning a common representation space to measure the similarity across the instances from different modalities. To our knowledge, none of the existing works is directly related to our approach that performs cross-modal retrieval of neural networks given datasets.

## 3 Methodology

We first define the Neural Network Search (NNS) problem and propose a meta-contrastive learning framework to learn a cross-modal retrieval space. We then describe the structural details of the query and model encoders, and an efficient model-zoo construction strategy.

### 3.1 Problem Definition

Our goal in NNS is to search for an optimal network (with both architectures and parameters) for a dataset and constraints, by learning a cross-modal latent space over the dataset-network pairs. We first describe the task-adaptive network retrieval with amortized meta-contrastive learning.

### 3.1.1 Meta-Training

We assume that we have a database of neural networks (model zoo) pretrained over a distribution of tasks $p(\tau)$ with each task $\tau = \{D^\tau, M^\tau, s^\tau\}$, where $D^\tau$ denotes a dataset, $M^\tau$ denotes a neural network (or model) trained on the dataset, and $s^\tau$ denotes a set of auxiliary constraints for the network to be found (e.g. number of parameters or the inference latency). Then, our goal is to learn a cross-modal latent space for the dataset-network pairs $(D^\tau, M^\tau)$ while considering the constraints $s^\tau$ over the task distribution $p(\tau)$, as illustrated in Figure 1. In this way, a meta-trained model from diverse $(D^\tau, M^\tau)$ pairs, will rapidly generalize on an unseen dataset $\tilde{D}$; $D^\tau \cap \tilde{D} = \emptyset$ by retrieving a well-fitted neural network on the dataset that satisfies the conditions $s^\tau$.

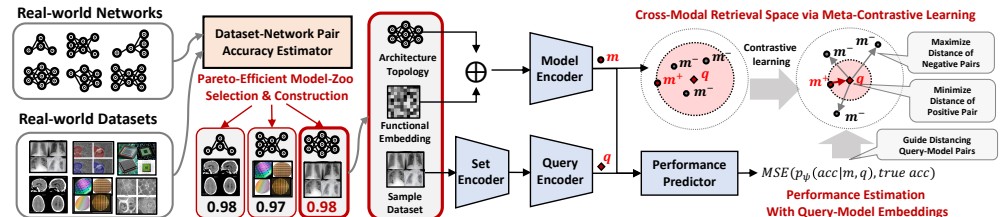

Figure 3: **Illustration for overall framework of our proposed method (TANS)**: We first construct our model-zoo with pareto-optimal pairs of dataset and network, rather than exhaustively train all possible pairs. We then embed a model and a dataset with a graph-functional model and a set encoder. After that, we meta-learn the cross-modal retrieval network over multiple model-query pairs, guided by our performance predictor.

**Task-Adaptive Neural Network Retrieval**    To learn a cross-modal latent space for dataset-network pairs over a task distribution, we first introduce a novel task-adaptive neural network retrieval problem. The goal of task-adaptive retrieval is to find an appropriate network $M^\tau$ given the query dataset $D^\tau$ for task $\tau$. To this end, we need to calculate the similarity between the dataset-network pair $(D^\tau, M^\tau) \in \mathcal{Q} \times \mathcal{M}$, with a scoring function $f$ that outputs the similarity between them as follows:

$$\max_{\boldsymbol{\theta}, \boldsymbol{\phi}} \sum_{\tau \in p(\tau)} f(\boldsymbol{q}, \boldsymbol{m}), \quad \boldsymbol{q} = E_Q(D^\tau; \boldsymbol{\theta}) \quad \text{and} \quad \boldsymbol{m} = E_M(M^\tau; \boldsymbol{\phi}), \tag{1}$$

where $E_Q : \mathcal{Q} \to \mathbb{R}^d$ is a query (dataset) encoder, $E_M : \mathcal{M} \to \mathbb{R}^d$ is a model encoder, which are parameterized with the parameter $\boldsymbol{\theta}$ and $\boldsymbol{\phi}$ respectively, and $f_{sim} : \mathbb{R}^d \times \mathbb{R}^d \to \mathbb{R}$ is a scoring function for the query-model pair. In this way, we can construct the cross-modal latent space for dataset-network pairs over the task distribution with equation 1, and use this space to rapidly retrieve the well-fitted neural network in response to the unseen query dataset.

We can learn such a cross-modal latent space of dataset-network pairs for rapid retrieval by directly solving for the above objective, with the assumption that we have the query and the model encoder: $Q$ and $M$. However, we further propose a contrastive loss to maximize the similarity between a dataset and a network that obtains high performance on it in the learned latent space, and minimize the similarity between irrelevant dataset-network pairs, inspired by Faghri et al. [19], Engilberge et al. [18]. While existing works such as Faghri et al. [19], Engilberge et al. [18] target image-to-text retrieval, we tackle the problem of cross-modal retrieval across datasets and networks, which is a nontrivial problem as it requires task-level meta-learning.

**Retrieval with Meta-Contrastive Learning**    Our *meta-contrastive learning* objective for each task $\tau \in p(\tau)$ consisting of a dataset-model pair $(D^\tau, M^\tau) \in \mathcal{Q} \times \mathcal{M}$, aims to maximize the similarity between positive pairs: $f_{sim}(\boldsymbol{q}, \boldsymbol{m}^+)$, while minimizing the similarity between negative pairs: $f_{sim}(\boldsymbol{q}, \boldsymbol{m}^-)$, where $\boldsymbol{m}^+$ is obtained from the sampled target task $\tau \in p(\tau)$ and $\boldsymbol{m}^-$ is obtained from other tasks $\gamma \in p(\tau); \gamma \neq \tau$, which is illustrated in Figure 3. This meta-contrastive learning loss can be formally defined as follows:

$$\mathcal{L}_m(\tau; \boldsymbol{\theta}, \boldsymbol{\phi}) = \mathcal{L}(f_{sim}(\boldsymbol{q}, \boldsymbol{m}^+), f_{sim}(\boldsymbol{q}, \boldsymbol{m}^-); \boldsymbol{\theta}, \boldsymbol{\phi}) \tag{2}$$
$$\boldsymbol{q} = E_Q(D^\tau; \boldsymbol{\theta}), \ \boldsymbol{m}^+ = E_M(M^\tau; \boldsymbol{\phi}), \ \boldsymbol{m}^- = E_M(M^\gamma; \boldsymbol{\phi}).$$

We then introduce $\mathcal{L}$ for the meta-contrastive learning:

$$\max \left( 0, \alpha - \log \frac{\exp(f_{sim}(\boldsymbol{q}, \boldsymbol{m}^+))}{\exp \left( \sum_{\gamma \in p(\tau), \gamma \neq \tau} f_{sim}(\boldsymbol{q}, \boldsymbol{m}^-) \right)} \right), \tag{3}$$

where $\alpha \in \mathbb{R}$ is a margin hyper-parameter and the score function $f_{sim}$ is the cosine similarity. The contrastive loss promotes the positive $(\boldsymbol{q}, \boldsymbol{m}^+)$ embedding pair to be close together, with at most margin $\alpha$ closer than the negative $(\boldsymbol{q}, \boldsymbol{m}^-)$ embedding pairs in the learned cross-modal metric space. Note that, similar to this, we also contrast the query with its corresponding model: $\mathcal{L}_q(f_{sim}(\boldsymbol{q}^+, \boldsymbol{m}), f_{sim}(\boldsymbol{q}^-, \boldsymbol{m}))$, which we describe in the supplementary material in detail.

With the above ingredients, we minimize the meta-contrastive learning loss over a task distribution $p(\tau)$, defined with the model $\mathcal{L}_m$ and query $\mathcal{L}_q$ contrastive losses, as follows:

$$\min_{\boldsymbol{\phi}, \boldsymbol{\theta}} \sum_{\tau \in p(\tau)} \mathcal{L}_m(\tau; \boldsymbol{\theta}, \boldsymbol{\phi}) + \mathcal{L}_q(\tau; \boldsymbol{\theta}, \boldsymbol{\phi}). \tag{4}$$

**Meta-Performance Surrogate Model**    We propose the meta-performance surrogate model to predict the performance on an unseen dataset without directly training on it, which is highly practical in real-world scenarios since it is expensive to iteratively train models for every dataset to measure the performance. Thus, we meta-train a performance surrogate model $a = S(\tau; \psi)$ over a distribution of tasks $p(\tau)$ on the model-zoo database. This model not only accurately predicts the performance $a$ of a network $M^\tau$ on an unseen dataset $D^\tau$, but also guides the learning of the cross-modal retrieval space, thus embedding a neural network closer to datasets that it performs well on.

Specifically, the proposed surrogate model $S$ takes a query embedding $\boldsymbol{q}^\tau$ and a model embedding $\boldsymbol{m}^\tau$ as an input for the given task $\tau$, and then forwards them to predict the accuracy of the model for the query. We train this performance predictor $S(\tau; \psi)$ to minimize the mean-square error loss $\mathcal{L}_s(\tau; \psi) = (s_{\text{acc}}^\tau - S(\tau; \psi))^2$ between the predicted accuracy $S(\tau; \psi)$ and the true accuracy $s_{\text{acc}}^\tau$ for the model on each task $\tau$, which is sampled from the task distribution $p(\tau)$. Then, we combine this objective with retrieval objective in equation 4 to train the entire framework as follows:

$$\min_{\boldsymbol{\phi},\boldsymbol{\theta},\boldsymbol{\psi}} \sum_{\tau \in p(\tau)} \mathcal{L}_m(\tau; \boldsymbol{\theta}, \boldsymbol{\phi}) + \mathcal{L}_q(\tau; \boldsymbol{\theta}, \boldsymbol{\phi}) + \lambda \cdot \mathcal{L}_s(\tau; \boldsymbol{\psi}), \tag{5}$$

where $\lambda$ is a hyper-parameter for weighting losses.

### 3.1.2 Meta-Test

By leveraging the meta-learned cross-modal retrieval space, we can instantly retrieve the best-fitted pretrained network $M \in \mathcal{M}$, given an unseen query dataset $\tilde{D} \in \tilde{\mathcal{Q}}$, which is disjoint from the meta-training dataset $D \in \mathcal{Q}$. Equipped with meta-training components described in the previous subsection, we now describe the details of our model at inference time, which includes the following: amortized inference, performance prediction, and task- and constraints-adaptive initialization.

**Amortized Inference**    Most existing NAS methods are slow as they require several GPU hours for training, to find the optimal architecture for a dataset $\tilde{D}$. Contrarily, the proposed *Task-Adaptive Network Search* (TANS) only requires a single forward pass per dataset, to obtain a query embedding $\tilde{\boldsymbol{q}}$ for the unseen dataset using the query encoder $Q(\tilde{D}; \boldsymbol{\theta}^*)$ with the meta-trained parameters $\boldsymbol{\theta}^*$, since we train our model with amortized meta-learning over a distribution of tasks $p(\tau)$. After obtaining the query embedding, we retrieve the best-fitted network $M^*$ for the query based on the similarity:

$$M^* = \max_{M^\tau}\{f_{sim}(\tilde{\boldsymbol{q}}, \boldsymbol{m}^\tau) \mid \tau \in p(\tau)\}, \tag{6}$$

where a set of model embeddings $\{\boldsymbol{m}^\tau \mid \tau \in p(\tau)\}$ is pre-computed by the meta-trained model encoder $E_M(M^\tau; \boldsymbol{\phi}^*)$.

**Performance Prediction**    While we achieve the competitive performance on unseen dataset only with the previously defined model, we also use the meta-learned performance predictor $S$ to select the best performing one among top $K$ candidate networks $\{\tilde{M}_i\}_{i=1}^K$ based on their predicted performances. Since this surrogate model with module to consider datasets is meta-learned over the distribution of tasks $p(\tau)$, we predict the performance on an unseen dataset $\tilde{D}$ without training on it. This is different from the conventional surrogate models [38, 75, 54] that additionally need to train on an unseen dataset from scratch, to predict the performance on it.

**Task-adaptive Initialization**    Given an unseen dataset, the proposed TANS can retrieve the network trained on a training dataset that is highly similar to the unseen query dataset from the model zoo (See Figure 4). Therefore, fine-tuning time of the trained network for the unseen target dataset $\tilde{D}$ is effectively reduced since the retrieved network $M$ has task-relevant initial parameters that are already trained on a similar dataset. If we need to further consider constraints $s$, such as parameters and FLOPs, then we can easily check if the retrieved models meet the specific constraints by sorting them in the descending order of their scores, and then selecting the constraint-fitted best accuracy model.

### 3.2 Encoding Datasets and Networks

**Query Encoder**    The goal of the proposed query encoder $E_Q(D; \boldsymbol{\theta}) : \mathcal{Q} \rightarrow \mathbb{R}^d$ is to embed a dataset $D$ as a single query vector $\boldsymbol{q}$ onto the cross-modal latent space. Since each dataset $D$ consists of $n$ data instances $D = \{X_i\}_{i=1}^n \in \mathcal{Q}$, we need to fulfill the permutation-invariance condition over

the data instances $X_i$, to output a consistent representation regardless of the order of its instances. To satisfy this condition, we first individually transform $n$ randomly sampled instances for the dataset $D$ with a continuous learnable function $\rho$, and then apply a pooling operation to obtain the query vector $\boldsymbol{q} = \sum_{X_i \in D} \rho(X_i)$, adopting Zaheer et al. [69].

**Model Encoder**    To encode a neural network $M^\tau$, we consider both its architecture and the model parameters trained on the dataset $D^\tau$ for each task $\tau$. Thus, we propose to generate a model embedding with two encoding functions: 1) topological encoding and 2) functional encoding.

Following Cai et al. [8], we first obtain a topological embedding $\boldsymbol{v}_t^\tau$ with auxiliary information about the architecture topology, such as the numbers of layers, channel expansion ratios, and kernel sizes. Then, our next goal is to encode the trained model parameters for the given task, to further consider parameters on the neural architecture. However, a major problem here is that directly encoding millions of parameters into a vector is highly challenging and inefficient. To this end, we use functional embedding, which embeds a network solely based on its input-output pairs. This operation generates the embedding of trained networks, by feeding a fixed Gaussian random noise into each network $M^\tau$, and then obtaining an output $\boldsymbol{v}_f^\tau$ of it. The intuition behind the functional embedding is straightforward: since networks with different architectures and parameters comprise different functions, we can produce different outputs for the same input.

With the two encoding functions, the proposed model encoder generates the model representation by concatenating the topology and function embeddings $[\boldsymbol{v}_t^\tau, \boldsymbol{v}_f^\tau]$, and then transforming the concatenated vector with a non-linear function $\sigma$ as follows: $\boldsymbol{m}^\tau = \sigma([\boldsymbol{v}_t^\tau, \boldsymbol{v}_f^\tau]) = E_M(M^\tau; \boldsymbol{\phi})$. Note that, the two encoding functions satisfy the injectiveness property under certain conditions, which helps with the accurate retrieval of the embedded elements in a condensed continuous latent space. We provide the proof of the injectiveness of the two encoding functions in Section C of the supplementary file.

### 3.3    Model-Zoo Construction

Given a set of datasets $\mathcal{D} = \{D_1, \ldots, D_K\}$ and a set of architectures $\mathcal{M} = \{M_1, \ldots, M_N\}$, the most straightforward way to construct a model zoo $\mathcal{Z}$, is by training all architectures on all datasets, which will yield a model zoo $\mathcal{Z}$ that contains $N \times K$ pretrained networks. However, we may further reduce the construction cost by collecting $P$ dataset-model pairs, $\{D, M\}_{i=1}^P$, where $P \ll N \times K$, by randomly sampling an architecture $M \in \mathcal{M}$ and then training it on $D \in \mathcal{D}$. Although this works well in practice (see Figure 8 (Bottom)), we further propose an efficient algorithm to construct it in a more sample-efficient manner, by skipping evaluation of dataset-model pairs that are certainly worse than others in all aspects (memory consumption, latency, and test accuracy). We start with an initial model zoo $\mathcal{Z}_{(0)}$ that contains a small amount of randomly selected pairs and its test accuracies. Then, at each iteration $t$, among the set of candidates $C_{(t)}$, we find a pair $\{D, M\}$ that can expand the currently known set of all Pareto-optimal pairs w.r.t. all conditions (memory, latency, and test accuracy on the dataset $D$), based on the amount of the Pareto front expansion estimated by $f_{zoo}(\cdot; \mathcal{Z}_{(t)})$:

$$\{D, M\}_{(t+1)} = \underset{(D, M) \in C_{(t)}}{\arg\max} \ f_{zoo}(\{D, M\}; \mathcal{Z}_{(t)}) \tag{7}$$

where $f_{zoo}(\{D, M\}; \mathcal{Z}) := \underset{\hat{s}_{acc}}{\mathbb{E}}[g_D(\mathcal{Z} \cup \{D, M, \hat{s}_{acc}\}) - g_D(\mathcal{Z})]$, $\hat{s}_{acc} \sim S(\{D, M\}; \boldsymbol{\psi}_{zoo})$ with parameter $\boldsymbol{\psi}_{zoo}$ trained on $\mathcal{Z}$, and the function $g_D$ measures the volume under the Pareto curve, also known as the Hypervolume Indicator [41], for the dataset $D$. We then train $M$ on $D$, and add it to the current model zoo $\mathcal{Z}_{(t)}$. For the full algorithm, please refer to Appendix A.

## 4    Experiments

In this section, we conduct extensive experimental validation against conventional NAS methods and commercially available AutoML platforms, to demonstrate the effectiveness of our proposed method.

### 4.1    Experimental Setup

**Datasets**    We collect **96** real-world image classification datasets from Kaggle*. Then we divide the datasets into two non-overlapping sets for **86** meta-training and **10** meta-test datasets. As some datasets contain relatively large number of classes than the other datasets, we adjust each dataset

---

*https://www.kaggle.com/

Table 1: **Performance of the searched networks on 10 unseen real-world datasets.** We report the averaged accuracy on ten unseen meta-test datasets over 3 different runs with 95% confidence intervals.

| Target Dataset | Method | # Epochs | FLOPs (M) | Params (M) | Search Time (GPU sec) | Training Time (GPU sec) | Speed Up | Accuracy (%) |
|---|---|---|---|---|---|---|---|---|
| **Averaged Performance** | MobileNetV3 [26] | 50 | 132.94 | 4.00 | - | $257.78_{\pm09.77}$ | 1.00× | $94.20_{\pm0.70}$ |
| | PC-DARTS [65] | 500 | 566.55 | **3.54** | $1100.37_{\pm22.20}$ | $5721.13_{\pm793.71}$ | 0.04× | $79.22_{\pm1.69}$ |
| | DrNAS [10] | 500 | 623.43 | 4.12 | $1501.75_{\pm43.92}$ | $5659.77_{\pm403.62}$ | 0.04× | $84.06_{\pm0.97}$ |
| | FBNet-A [60] | 50 | 246.69 | 4.3 | - | $293.42_{\pm57.45}$ | 0.88× | $93.00_{\pm1.95}$ |
| | OFA [8] | 50 | 148.76 | 6.74 | $121.90_{\pm0.00}$ | $226.58_{\pm03.13}$ | 0.74× | $93.89_{\pm0.84}$ |
| | MetaD2A [31] | 50 | 512.67 | 6.56 | $2.59_{\pm0.13}$ | $345.39_{\pm28.36}$ | 0.74× | $95.24_{\pm1.14}$ |
| | **TANS (Ours)** | 10 | 181.74 | 5.51 | $0.002_{\pm0.00}$ | $40.19_{\pm03.06}$ | - | $95.17_{\pm2.20}$ |
| | **TANS (Ours)** | 50 | 181.74 | 5.51 | $\mathbf{0.002_{\pm0.00}}$ | $200.93_{\pm11.01}$ | **1.28×** | $\mathbf{96.28_{\pm0.30}}$ |
| Colorectal Histology Dataset (Easy) | MobileNetV3 [26] | 50 | 132.94 | **4.00** | - | $577.18_{\pm04.15}$ | 1.00× | $96.23_{\pm0.07}$ |
| | PC-DARTS [65] | 500 | 534.64 | 4.02 | $2062.42_{\pm49.14}$ | $12124.18_{\pm1051.16}$ | 0.04× | $96.17_{\pm0.68}$ |
| | DrNAS [10] | 500 | 614.23 | 4.12 | $4183.20_{\pm188.60}$ | $11355.18_{\pm1352.62}$ | 0.04× | $97.51_{\pm0.13}$ |
| | FBNet-A [60] | 50 | 215.45 | 4.3 | - | $696.00_{\pm295.19}$ | 0.83× | $95.43_{\pm0.57}$ |
| | OFA [8] | 50 | 134.85 | 6.74 | $121.90_{\pm0.00}$ | $537.61_{\pm03.52}$ | 0.88× | $96.40_{\pm0.52}$ |
| | MetaD2A [31] | 50 | 506.88 | 5.93 | $2.58_{\pm0.12}$ | $784.45_{\pm79.32}$ | 0.73× | $96.57_{\pm0.56}$ |
| | **TANS (Ours)** | 10 | 171.74 | 4.95 | $0.001_{\pm0.00}$ | $98.56_{\pm04.24}$ | - | $96.87_{\pm0.21}$ |
| | **TANS (Ours)** | 50 | 171.74 | 4.95 | $\mathbf{0.001_{\pm0.00}}$ | $492.81_{\pm21.19}$ | **1.17×** | $\mathbf{97.67_{\pm0.05}}$ |
| Food Classification Dataset (Hard) | MobileNetV3 [26] | 50 | 132.94 | **4.00** | - | $235.57_{\pm07.57}$ | 1.00× | $87.52_{\pm0.78}$ |
| | PC-DARTS [65] | 500 | 567.85 | **3.62** | $1018.49_{\pm6.31}$ | $6323.40_{\pm938.83}$ | 0.03× | $55.42_{\pm2.46}$ |
| | DrNAS [10] | 500 | 632.67 | 4.12 | $1276.38_{\pm0.00}$ | $5079.89_{\pm161.05}$ | 0.04× | $61.45_{\pm0.68}$ |
| | FBNet-A [60] | 50 | 251.29 | 4.3 | - | $251.24_{\pm3.31}$ | 0.94× | $84.33_{\pm1.41}$ |
| | OFA [8] | 50 | 152.34 | 6.74 | $121.90_{\pm0.00}$ | $\mathbf{190.86_{\pm03.48}}$ | 0.75× | $87.43_{\pm0.59}$ |
| | MetaD2A [31] | 50 | 521.11 | 8.23 | $2.60_{\pm0.23}$ | $324.62_{\pm34.97}$ | 0.72× | $89.72_{\pm1.53}$ |
| | **TANS (Ours)** | 10 | 179.83 | 5.07 | $0.002_{\pm0.00}$ | $40.59_{\pm04.84}$ | - | $93.11_{\pm0.24}$ |
| | **TANS (Ours)** | 50 | 179.83 | 5.07 | $\mathbf{0.002_{\pm0.00}}$ | $202.93_{\pm24.21}$ | **1.16×** | $\mathbf{93.71_{\pm0.24}}$ |

| Query | Retrieval | Query | Retrieval | Query | Retrieval |
|---|---|---|---|---|---|

Figure 4: **Retrieved examples for meta-test datasets.** We visualize the dataset used for pretraining the retrieved model with the unseen query dataset. For more examples, see Figure 9 in Appendix.

to have up to 20 classes, yielding 140 and 10 datasets for meta-training and meta-test datasets, respectively (Please see Table 5 for detailed dataset configuration). For each dataset, we use randomly sampled 80/20% instances as a training and test set. To be specific, our 10 meta-test datasets include Colorectal Histology, Drawing, Dessert, Chinese Characters, Speed Limit Signs, Alien vs Predator, COVID-19, Gemstones, and Dog Breeds. We strictly verified that there is **no dataset-, class-, and instance-level overlap** between the meta-training and the meta-test datasets, while some datasets may contain semantically similar classes.

**Baseline Models** We consider MobileNetV3 [26] pretrained on ImageNet as our baseline neural network. We compare our method with conventional NAS methods, such as PC-DARTS [65] and DrNAS [10], weight-sharing approaches, such as FBNet [60] and Once-For-All [8], and data-driven meta-NAS approach, e.g. MetaD2A [31]. All these NAS baseline approaches are based on MobileNetV3 pretrained on ImageNet, except for the conventional NAS methods that are only able to generate architectures. As such conventional NAS methods start from the scratch, we train them for sufficient amount of training epochs (10 times more training steps) for fair comparison. Please see Appendix B for further detailed descriptions of the experimental setup.

**Model-zoo Construction** We follow the OFA search space [8], which allows us to design resource-efficient architectures, and thus we obtain network candidates from the OFA space. We sample 100 networks condidates per meta-training dataset, and then train the network-dataset pairs, yielding 14,000 dataset-network pairs in our full model-zoo. We also construct smaller-sized efficient model-zoos from the full model-zoo (14,000) with our efficient algorithm described in Section 3.3. We use the full-sized model-zoo as our base model-zoo, unless otherwise we clearly mention the size of the used model-zoo. Detailed descriptions, e.g. training details and costs, are described in Appendix D.2.

## 4.2 Experimental Results

**Meta-test Results** We compare 50-epoch accuracy between TANS and the existing NAS methods on 10 novel real-world datasets. For a fair comparison, we train PC-DARTS [65] and DrNAS [10] for 10 times more epochs (500), as they only generate architectures without pretrained weights, so we train them for a sufficient amount of iterations. For FBNet and OFA (weight-sharing methods) and

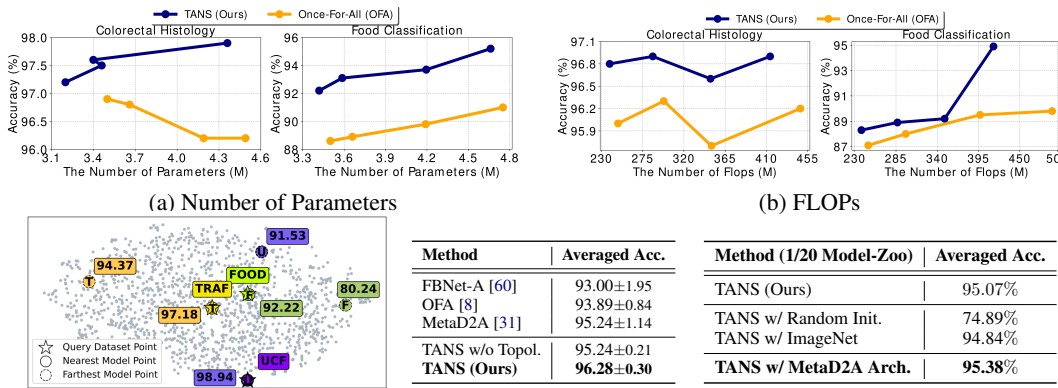

(a) Number of Parameters  (b) FLOPs

| Method | Averaged Acc. |
|---|---|
| FBNet-A [60] | 93.00±1.95 |
| OFA [8] | 93.89±0.84 |
| MetaD2A [31] | 95.24±1.14 |
| TANS w/o Topol. | 95.24±0.21 |
| **TANS (Ours)** | **96.28±0.30** |

| Method (1/20 Model-Zoo) | Averaged Acc. |
|---|---|
| TANS (Ours) | 95.07% |
| TANS w/ Random Init. | 74.89% |
| TANS w/ ImageNet | 94.84% |
| **TANS w/ MetaD2A Arch.** | **95.38%** |

(c) The Cross-Modal Latent Space   (d) Effectiveness of Topology   (e) Analysis on Arch. & Parameters

Figure 6: **In-depth analysis of TANS:** Performance comparison with constraints, such as (a) the number of parameters and (b) FLOPs. (c) Visualization for the cross-modal latent space using T-SNE and performance comparison depending on the distance (d) Ablation study on topology information. (e) Analysis on the architecture and pretrained knowledge.

MetaD2A (data-driven meta-NAS), which provide the searched architectures as well as pretrained weights from ImageNet-1K, we fine-tune them on the meta-test query datasets for 50 epochs.

As shown in Table 1, we observe that TANS out-performs all baselines, with incomparably smaller search time and relatively smaller training time. Conventional NAS approaches such as PC-DARTS and DrNAS repeatedly require large search time for every dataset, and thus are inefficient for this practical setting with real-world datasets. FBNet, OFA, and MetaD2A are much more efficient than

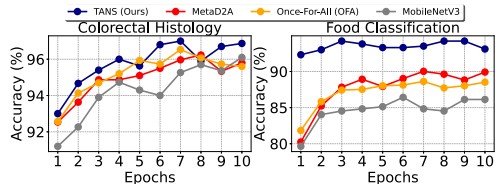

Figure 5: Meta-test Accuracy Curves

general NAS methods since they search for subnets within a given supernet, but obtain suboptimal performances on unseen real-world datasets as they may have largely different distributions from the dataset the supernet is trained on. In contrast, our method achieves almost zero cost in search time, and reduced training time as it fine-tunes a network pretrained on a relevant dataset. In Figure 5, we show the test performance curves and observe that TANS often starts with a higher starting point, and converges faster to higher accuracy.

In Figure 4, we show example images from the query and training datasets that the retrieved models are pretrained on. In most cases, our method matches semantically similar datasets to the query dataset. Even for the semantically-dissimilar cases (right column), for which our model-zoo does not contain models pretrained on datasets similar to the query, our models still outperform all other base NAS models. As such, our model effectively retrieves not only task-relevant models, but also potentially best-fitted models even trained with dissimilar datasets, for the given query datasets. We provide detailed descriptions for all query and retrieval pairs in Figure 9 of Appendix.

We also compare with commercially available AutoML platforms, such as Microsoft Azure Custom Vision [1] and Google Vision Edge [2]. For this experiment, we evaluate on randomly chosen five datasets (out of ten), due to excessive training costs required on AutoML platforms. As shown in Figure 2, our method outperforms all commercial NAS/AutoML methods, with a significantly smaller total time cost. We provide more details and additional experiments, such as including real-world architectures, in Appendix E.

**Analysis of the Architecture & Parameters**   To see that our method is effective in retrieving networks with both optimal architecture and relevant parameters, we conduct several ablation studies. We first report the results of base models that only search for the optimal architectures. Then we provide the results of the network retrieved using a variant of our method which does not use the topology (architecture) embedding, and only uses the functional embedding $v_f^\tau$ (Tans w/o Topol). As shown in Figure 6 (d), TANS w/o Topol outperforms base NAS methods (except for MetaD2A) without considering the architectures, which shows that the performance improvement mostly comes from the knowledge transfer from the weights of the most relevant pretrained networks. However, the full TANS obtains about 1% higher performance over TANS w/o Topol., which shows the importance

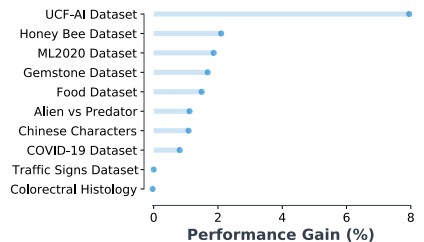

(a) Effectiveness of Performance Predictor

| Model | Performance of The Same Pair Retrieval | | | | |
|---|---|---|---|---|---|
| | R@1 | R@5 | R@10 | Mean | Median |
| Random | 2.14 | 2.86 | 8.57 | 69.04 | 70.0 |
| Largest Parameter | 3.57 | 7.14 | 10.00 | 51.85 | 52.0 |
| TANS + Cosine | 9.29 | 12.86 | 22.14 | 46.02 | 38.0 |
| TANS + Hard Neg. | 72.14 | 84.29 | 88.57 | 4.86 | 1.0 |
| **TANS + Contrastive** | **80.71** | **96.43** | **99.29** | **1.90** | **1.0** |
| TANS w/o Func.Emb. | 5.00 | 11,43 | 18.57 | 63.20 | 63.0 |
| TANS w/o Predictor | 80.00 | **96.43** | 97.86 | 2.23 | **1.0** |

(b) Analysis of Retrieval Performance

| Method | MSE on 5 Meta-test Datasets | | | | |
|---|---|---|---|---|---|
| | Food | Gemstones | Dogs | A. vs P. | COVID-19 |
| Predictor w/o $v_q^\tau$ | 0.0178 | 0.0782 | 0.0194 | 0.0185 | 0.0418 |
| Predictor w/o $v_f^\tau$ | 0.0188 | **0.0323** | 0.0016 | 0.0652 | 0.0328 |
| **Predictor (Ours)** | **0.0036** | 0.0338 | **0.0013** | **0.0028** | **0.0233** |

(c) Analysis of Performance Predictor

| Method | MSE on 5 Meta-test Datasets | | | | |
|---|---|---|---|---|---|
| | Food | Gemstones | Dogs | A. vs P. | COVID-19 |
| Random Estimation | 0.1619 | 0.1081 | 0.1348 | 0.2609 | 0.2928 |
| 1/100 Model-Zoo | 0.0088 | 0.0369 | 0.0034 | 0.0077 | 0.0241 |
| **Top 10 Retrievals** | **0.0036** | **0.0338** | **0.0013** | **0.0028** | **0.0233** |

(d) Effectiveness of Performance Predictor

Figure 7: **In-depth analysis of TANS (2):** (a) The effectiveness of meta-performance predictor. (b) Ablation study on retrieval performance. Additional (c) analysis and (d) the effectiveness of our performance predictor.

of the architecture and the effectiveness of our architecture embedding. In Figure 6 (e), we further experiment with a version of our method that initializes the retrieved networks with both random weights and ImageNet pre-trained weights, using 1/20 sized model-zoo (700). We observe that they achieve lower accuracy over TANS on 10 datasets, which again shows the importance of retrieving relevant tasks' knowledge. We also construct the model-zoo by training on the architectures found by an existing NAS method (MetaD2A), and see that it further improves the performance of TANS.

**Contraints-conditioned Retrieval**    TANS can retrieve models with a given dataset and additional constraints, such as the number of the parameters or the computations (in FLOP). This is practically important since we may need a network with less memory and computation overhead depending on the hardware device. This can be done by filtering networks that satisfy the given conditions among the candidate networks retrieved. For this experiment, we compare against OFA that performs the same constrained search, as other baselines do not straightforwardly handle this scenario. As shown in Figure 6 (a) and (b), we observe that the network retrieved with TANS consistently outperforms the network searched with OFA under varying parameters and computations constraints. Such constrained search is straightforward with our method since our retrieval-based method allows us to search from the database consisting of networks with varying architectures and sizes.

**Analysis of the Cross-Modal Retrieval Space**    We further examine the learned cross-modal space. We first visualize the meta-learned latent space in Figure 6 (c) with randomly sampled 1,400 models among 14,000 models in the model-zoo. We observe that the network whose embeddings are the closest to the query dataset achieves higher performance on it, compared to networks embedded the farthest from it. For example, the accuracy of the closet network point for UCF-AI is 98.94% while the farthest network only achieves 91.53%.

We also show Spearman correlation scores on 5 meta-test datasets in Table 2. Measuring correlation with the distances is not directly compatible with our contrastive loss, since the negative examples (models that achieve low performance on the target

Table 2: Latent Distance and Performance

| Spearman's Correlation on 5 Meta-test Datasets | | | | |
|---|---|---|---|---|
| Food | Drawing | Chinese | A. vs P. | Colorectal |
| 0.752 | 0.583 | 0.322 | 0.214 | 0.213 |

dataset) are pushed away from the query point, without a meaningful ranking between the negative instances. To obtain a latent space where the negative examples are also well-ranked, we should replace the contrastive loss with a ranking loss instead, but this will not be very meaningful. Hence, to effectively validate our latent space with correlation metric, we rather select clusters, such that 50 models around the query point and another 50 models around the farthest points, thus using a total of 100 models to report the correlation scores. In the Table 2, we show the correlation scores of these 100 models on the five unseen datasets. For Food dataset (reported as "hard" dataset in Table 1), the correlation scores are shown to be high. On the other hand, for Colorectal Histology dataset (reported as "easy" dataset), the correlation scores are low as any model can obtain good performance on it, which makes the performance gap across models small. In sum, as the task (dataset) becomes more difficult, we can observe a higher correlation in the latent space between the distance and the rank.

**Meta-performance Predictor** The role of the proposed performance predictor is not only guiding the model and query encoder to learn the better latent space but also selecting the best model among retrieval candidates. To verify its effectiveness, we measure the performance gap between the top-1 retrieved model w/o the predictor and the model with the highest scores selected using the predictor among retrieval candidates. As shown in Figure 7 (a), there are $1.5\%p$ - $8\%p$ performance gains on the meta-test datasets. The top-1 retrieved model from the model zoo with TANS may not be always optimal for an unseen dataset, and our performance predictor remedies this issue by selecting the best-fitted model based on its estimation. We also examine ablation study for our performance predictor. Please note that we do not use ranking loss which does not rank the negative examples. Thus we use Mean Squared Error (MSE) scores. We retrieve the top 10 most relevant models for an unseen query datasets and then compute the MSE between the estimated scores and the actual ground truth accuracies. As shown in Figure 7 (c), we observe that removing either query or model embeddings degrades performance compared to the predictor taking both embeddings. It is natural that, with only model or query information, it is difficult to correctly estimate the accuracy since the predictor fails to recognize what or where to learn. Also, we report the MSE between the predicted performance using the predictor and the ground truth performance of each model for the entire set of pretrained models from a smaller model zoo in Figure 7 (d). Although the performance predictor achieves slightly higher MSE scores for this experiment compared to the MSE obtained on the top-10 retrieved models (which are the most important), the MSE scores are still meaningfully low, which implies that our performance model successfully works even toward the entire model-zoo.

**Retrieval Performance** We also verify whether our model successfully retrieves the same paired models when the correspondent **meta-train** datasets are given (we use unseen instances that are **not used** when training the encoders.) For the evaluation metric, we use recall at $k$ (**R@$k$**) which indicates the percentage of the correct models retrieved among the top-$k$ candidates for the unseen query instances, where $k$ is set to 1, 5, and 10. Also, we report the mean and median ranking of the correct network among all networks for the unseen query. In Figure 7 (b), the largest parameter selection strategy shows poor performances on the retrieval problem, suggesting that simply selecting the largest network is not a suitable choice for real-world tasks. In addition, compared to cosine similarity learning, the proposed meta-contrastive learning allows the model to learn significantly improved discriminative latent space for cross-modal retrieval. Moreover, without our performance predictor, we observe that TANS achieves slightly lower performance, while it is significantly degenerated when training without functional embeddings.

**Analysis of Model-Zoo Construction** Unlike most existing NAS methods, which repeatedly search the optimal architectures *per dataset* from their search spaces, TANS do not need to perform such repetitive search procedure, once the model-zoo is built beforehand. We are able to adaptively retrieve the relevant pretrained models for **any number of datasets** from our model-zoo, with almost zero search costs. Formally, TANS reduces the time complexities of both search cost and pre-training cost **from O(N) to O(1)**, where $N$ is the number of datasets, as shown in Figure 8 (Top). Furthermore, a model zoo constructed using our efficient construction algorithm, introdcued in Section 3.3, yields models with higher performance on average, compared to the random sampling strategy when the size of the model-zoo is the same as shown in Figure 8 (Bottom).

| Method | Architecture Search Cost | Retrieved Dataset Pre-training Cost |
|---|---|---|
| Conventional NAS | O(N) | O(N) |
| MetaD2A | O(1) | O(N) |
| **TANS (Ours)** | **O(1)** | **O(1)** |

Figure 8: Search & Pretraining Costs (Top), Model-Zoo Analysis (Bottom)

# 5 Conclusion

We propose a novel Task-Adaptive Neural Network Search framework (TANS), that instantly retrieves a relevant pre-trained network for an unseen dataset, based on the cross-modal retrieval of dataset-network pairs. We train this retrieval model via amortized meta-learning of the cross-modal latent space with contrastive learning, to maximize the similarity between the positive dataset-network pairs, and minimize the similarity between the negative pairs. We train our framework on a model zoo consisting of diverse pretrained networks, and validate its performance on ten unseen datstes. The results show that the proposed TANS rapidly searches and trains a well-fitted network for unseen datasets with almost no architecture search cost and significantly fewer fine-tuning steps to reach the target performance, compared to other NAS methods. We discuss the **limitation and the societal impact** of our work in Appendix F.

**Acknowledgement** This work was supported by AITRICS and by Institute of Information & communications Technology Planning & Evaluation (IITP) grant funded by the Korea government(MSIT) (No.2019-0-00075, Artificial Intelligence Graduate School Program(KAIST))

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
