# Appendix

**Organization**    In Appendix, we provide detailed descriptions of the materials that are not fully covered in the main paper, and provide additional experimental results, which are organized as follows:

- **Section A** - We describe the *implementation details* of our model-zoo construction, query and model encoders, and meta-surrogate performance predictor.
- **Section B** - We provide the details of the *model training*, such as the learning rate and hyper-parameters, of meta-train/test and constructing model-zoo.
- **Section C** - We provide the *proof of injectiveness* with the proposed query and model encoding functions over the cross-modal latent space.
- **Section D** - We elaborate on the detailed *experimental setups*, such as architecture space, baselines, and datasets, corresponding to the experiments introduced in the main document.
- **Section E** - We provide additional *analysis* of the experiments introduced in the main document and present the experiment with different model-zoo settings.
- **Section F** - We discuss *the societal impact and the limitation* of our work.

## A    Implementation Details

### A.1    Efficient Model Zoo Construction

---
**Algorithm 1:** Model Zoo Construction

---
**Input**    : $\mathcal{D}, \mathcal{M}$: collection of datasets and models, respectively,
$\qquad\quad \mathcal{Z}_{(0)} \subset \mathcal{D} \times \mathcal{M} \times \mathbb{R}_{[0,1]}$: set of $N_{init}$ initial tuples of (dataset, model, test accuracy)
$t \leftarrow 0$;
**while** *termination condition is not met* **do**
$\qquad$ **if** *t is divisible by $N_{train}$* **then**
$\qquad\qquad$ Train accuracy predictor parameters $\psi_{zoo}$ on data $\mathcal{Z}_{(t)}$;
$\qquad$ $C_{(t)} \leftarrow$ Choose a subset of candidate pairs from $\mathcal{D} \times \mathcal{M}$ not present in $\mathcal{Z}_{(t)}$;
$\qquad$ $(D, M) \leftarrow \max_{(D,M) \in C_{(t)}} f_{zoo}(C; \mathcal{Z}_{(t)})$;
$\qquad$ $\alpha^* \leftarrow$ Evaluate the actual accuracy of $(D, M)$ by training $M$ on $D$;
$\qquad$ $\mathcal{Z}_{(t+1)} \leftarrow \mathcal{Z}_{(t)} \cup (D, M, \alpha^*)$;
$\qquad$ $t \leftarrow t + 1$;
**Output :** Efficiently constructed model zoo $\mathcal{Z}_{(t)}$.

---

The algorithm that is used to efficiently construct the model-zoo is described in Algorithm 1. The score function $f_{zoo}(D, M; \mathcal{Z})$, which measures how much adding a pair $(D, M)$ will improve upon the model zoo $\mathcal{Z}$, is defined as

$$f_{zoo}(D, M; \mathcal{Z}) := \mathbb{E}_{\hat{s}_{acc} \sim S((D,M); \psi_{zoo})}[g_D(\mathcal{Z} \cup (D, M, \hat{s}_{acc})) - g_D(\mathcal{Z})] \qquad (8)$$

where $S$ indicates the accuracy predictor, and $g_D$ is the normalized volume under the pareto-dominated pairs for dataset $D$:

$$g_D \left( \left\{ (D^{(i)}, M^{(i)}, s_{acc}^{(i)}) \right\}_{i=1}^n \right) \qquad (9)$$

$$:= \text{Hypervolume} \left( \left\{ \left( s_{acc}^{(i)}, \tilde{s}_{latency}(M^{(i)}), \tilde{s}_{parameters}(M^{(i)}) \right) \Big| D^{(i)} = D \right\} \right) \qquad (10)$$

where $\tilde{s}_{latency}(M)$ and $\tilde{s}_{parameters}(M)$ indicates the normalized latency and the normalized number of parameters of the model $M$, respectively. The latency and parameters are normalized so that the maximum value across all models becomes 1.0 and the minimum value becomes 0.0. The hypervolume can be computed efficiently with the `PyGMO` library [5].

The accuracy predictor used in the model-zoo construction is very similar to the structure of the performance predictor described in Section A.4, but we used a functional embedding obtained from the model pretrained on Imagenet-1K, instead of a functional embedding obtained from a model already trained on the target dataset, since training the model on the target dataset just to obtain the functional embedding would defeat the purpose of this algorithm. Also, to incorporate uncertainty about the accuracy predictions, we use 10 samples from the accuracy predictor with MC dropout to evaluate the expectation in (8). The dropout probability is set to 0.5.

### A.2    Query Encoder

Our query encoder takes sampled instances (e.g. 10 unseen random images per class) as an input from the query dataset. We use image embeddings from ResNet18 [23] pretrained on ImageNet 1K [13], whose dimensions

are 512 (except for the last classification layer), rather than using raw images, simply to reduce computation costs. We then use a linear layer with 512 dimensions, followed by $Mean$ pooling and $L_2$ normalization, which outputs encoded vectors with 128 dimensions. As we use Deep set [69], we tried performing $Sum$ pooling, instead of $Mean$ pooling, however, we observe that taking the average on instances shows better R@$k$ scores for the correct pair retrieval, and thus we use $Mean$ pooling when encoding query samples.

### A.3 Model Encoder

Our model encoder takes both OFA flat topology [8] and functional embedding as an input. For the flat topology, it contains information such as kernel size, width expansion ratio, and depth, in a 45-dimensional vector. In addition, the functional embedding, which bypasses the need for direct parameter encoding, represents models' learned knowledge in a 1536-dimensional vector. We first concatenate both vectors and normalize the vector. Then, we learn a projection layer, which is a 1581-length fully-connected layer, followed by $L_2$ norm operation, which outputs encoded vectors with 128 dimensions.

### A.4 Meta-Surrogate Performance Predictor

Our performance predictor takes both query and model embeddings simultaneously. Both embedding vectors are 128-dimensional vectors. We first concatenate the embeddings into 256-dimensional vectors and then forward them through a 256-length fully connected layer. We then produce a single continuous value for a predicted accuracy. We perform a sigmoid operation on the values to range the values into a scale from 0.0 to 1.0.

## B  Training Details

There are two steps of training required for our Task-Adaptive Network Search (TANS): 1) training the cross-modal latent space and 2) fine-tuning the retrieved model on an unseen meta-test dataset.

### B.1 Learning the Cross-modal Latent Space

For the model-zoo encoding, we set the batch size to 140 as we have 140 different datasets. Since, for each dataset, we randomly choose one model among 100 models from each dataset. Then we minimize the contrastive loss on the 140 samples. Although we train our encoders over a large number of dataset-network pairs (14,000 models), the entire training time takes less than two hours based on NVIDIA's RTX 2080 TI GPU. We initialize our model weights with the same value across all encoders and experiments, rather than differently initializing the encoders for every experimental trial. We use the Adam optimizer (We use the learning rate of 1e-2).

### B.2 Fine-tuning on Meta-test Datasets

For the fine-tuning phase, we set all settings, such as hyper-parameters, learning rate, optimizer, etc., exactly the same across all baseline models and our method, and the differences are clearly mentioned in this section otherwise. We use the SGD optimizer with an initial learning rate (1e-2), weight decay (4e-5), and momentum (0.9). Also, we use the *Cosine Annealing* learning rate scheduler. We train the models with 224 sized images (after resizing) and we set batch-size to 32, except PC-DARTS which has memory issues with 224 sized images (for PC-DARTS, we set to 12 as the batch-size), and DrNAS which we train with 32 sized images due to *heavy* training time costs. We train all models for 50 epochs and we show that our model converges faster than all baseline models.

### B.3 Constructing the Model-Zoo

For the model-zoo consisting of 14,000 random pairs used in the main experiment, we fine-tune the ImageNet1K pretrained OFA models on the dataset for 625 epochs, following the progressive shrinking method described in [8]. We then choose 100 random OFA architectures for each dataset and evaluate their test accuracies on the test split.

For the efficiently constructed model zoo experiment, we use the algorithm described in Section 3.3 and further elaborated in Section A.1, using the 14,000-pair model zoo as the search space. For the initial samples, we use $N_{init} = 750$, where 5-6 samples are taken from each dataset. The accuracy predictor is retrained from scratch every 64 iterations until the validation accuracy no longer improves for 5 epochs.

## C  Proof for Uniqueness of the Query and Model Encoding Functions

In this section, we show that the proposed query and model encoders can represent the injective function on the input query $D \in \mathcal{Q}$ and model $M \in \mathcal{M}$, respectively.

**Proposition 1 (Injectiveness on Query Encoding).** *Assume $\mathcal{Q}$ and $D$ are finite sets. A query encoder $E_Q$ : $\mathcal{Q} \to \mathbb{R}^d$ can injectively map two different queries $D_1, D_2$ into distinct embeddings $\boldsymbol{q_1}, \boldsymbol{q_2}$, where $D \in \mathcal{Q}$ and $\boldsymbol{q} \in \mathbb{R}^d$.*

*Proof.* A query encoder $E_Q$ maps a query dataset $D \in \mathcal{Q}$ to a vector $\boldsymbol{q} \in \mathbb{R}^d$ as follows: $E_Q : D \mapsto \boldsymbol{q}$, where $\mathcal{Q}$ is a set of queries, which contains a set of data instances $X$ for constructing a dataset $D = \{X_1, X_2, ..., X_n\}$. Then, our goal here is to make a query encoder that uniquely maps two different queries $D_1, D_2$ into two distinct embeddings $\boldsymbol{q}_1, \boldsymbol{q}_2$.

Each dataset $D$ consists of $n$ data instances: $D = \{X_1, X_2, ..., X_n\}$, where $n$ is smaller than the number of elements in $\mathbb{N}$. To encode each query dataset $D$ into a vector space, as described in query encoder paragraph of section 3.2, we first transform each instance $X_i$ into the representation space with a continuous function $\rho$, and then aggregate all set elements, which is adopted from Zaheer et al. [69]. In other words, a query encoder can be defined as follows: $\boldsymbol{q} = \sum_{X_i \in D} \rho(X_i)$.

We assume that $\mathcal{Q}$ is a finite set, and each $D = \{X_1, X_2, ..., X_n\} \in \mathcal{Q}$ is also a finite set with $|D| = n$ elements. Therefore, a set of data instances $X$ is countable, since the product of two nature numbers (i.e. $|\mathcal{Q}| \times n$) is a natural number. For this reason, there can be a unique mapping $Z$ from the element $X$ to the nature number in $\mathbb{N}$. If we let $\rho(X) = 4^{-Z(X)}$, then the form of a query encoder $\sum_{X_i \in D} \rho(X_i)$ constitutes an unique mapping for every set $D \in \mathcal{Q}$ (see Zaheer et al. [69], Wagstaff et al. [57] for details). In other words, the output of the query encoder is unique for each input dataset $D$ that consists of $n$ data instances. $\qquad\square$

Thanks to the universal approximation theorem [25, 24], we can construct a mapping function $\rho$ using multi-layer perceptrons (MLPs).

**Proposition 2 (Injectiveness on Model Encoding).** *Assume $\mathcal{M}$ is a countable set. A model encoder $E_M$ : $\mathcal{M} \to \mathbb{R}^d$ can injectively map two different architectures $M_1, M_2$ into distinct embeddings $\boldsymbol{m}_1, \boldsymbol{m}_2$, where $M \in \mathcal{M}$ and $\boldsymbol{m} \in \mathbb{R}^d$.*

*Proof.* As described in the model encoder paragraph of Section 3.2, we represent each neural network $M \in \mathcal{M}$ with both topological embedding $\boldsymbol{v}_t$ and functional embedding $\boldsymbol{v}_f$. Thus, if one of two embeddings can uniquely represent each neural network, then the injectiveness on model encoding $E_M : \mathcal{M} \to \mathbb{R}^d$ is satisfied.

We first show that the topological encoding function $E_{M_T} : M \mapsto \boldsymbol{v}_t$ can uniquely represent each architecture $M$ in the embedding space. As described in the Model Encoder part of section A, we use a 45-dimensional vector that contains topological information, such as the number of layers, channel expansion ratios, and kernel sizes (See Cai et al. [8] for details), for the topological encoding. Also, each topological information uniquely defines each neural architecture. Therefore, the embedding $\boldsymbol{v}_t$ from the topological encoding function $E_{M_T}$ is unique on each neural network $M$.

While we can obtain the distinct embedding of each neural network with the topological encoding function alone, we also consider the injectiveness of the functional encoding in the following. To consider the functional embedding, we first model a neural architecture as its computational graph, which can be further denoted as a directed acyclic graph (DAG). Using this computational graph scheme, a functional model encoder $E_{M_F}$ maps an architecture (computational graph) $M \in \mathcal{M}$ into a vector $\boldsymbol{v}_f$ as follows: $E_{M_F} : M \mapsto \boldsymbol{v}_f$. Then, our goal here is to make the functional encoder $E_{M_F}$ that uniquely maps two different neural architectures $M_1, M_2$ into two different embeddings $\boldsymbol{v}_{f_1}, \boldsymbol{v}_{f_2}$, with the computational graph represented as the DAG structure.

Assume that a computational graph for a neural network $M$ has $n$ nodes. Then, each node $v_i$ on the graph has its corresponding operation $o_i$, which transforms incoming features for the node $v_i$ to an output representation $C_i$. In other words, $C_i$ indicates the output of the composition of all operations along the path from $v_1$ to $v_i$.

Particularly, in our model encoder case, we have an arbitrary input signal $\boldsymbol{x}$ that is the fixed Gaussian noise, where we insert this fixed input into the starting node $v_1$ (See Model Encoder paragraph of section 3.2 for details). Also, for the simplicity of the notation, we set $C_0 = \boldsymbol{x}$ that is the output of the virtual node $v_0$ and the incoming representation of the starting node of the computational graph. Then, the output representation for the node $v_i$ is formally defined as follows: $C_i(\boldsymbol{x}) = o_i(\{C_j(\boldsymbol{x}) : v_j \to v_i\})$, where $\{C_j(\boldsymbol{x}) : v_j \to v_i\}$ denotes a multiset for the output representation of $v_i$'s predecessors, and the operation $o_i$ transforms the incoming representations over the multiset into the output representation. Note that, to consider the multiplicity of the nodes on a graph, we use a multiset scheme, rather than a set [64].

Following the proof of Theorem 2 in Zhang et al. [72], we rewrite the $C_i(\boldsymbol{x}) = o_i(\{C_j(\boldsymbol{x}) : v_j \to v_i\})$ as follows: $C_i(\boldsymbol{x}) = \omega(o_i, \{C_j(\boldsymbol{x}) : v_j \to v_i\})$, where $\omega$ is an injective function over two inputs $o_i$ and $\{C_j(\boldsymbol{x}) : v_j \to v_i\}$. Then, $C_i$ can uniquely embed the output representation of node $v_i$, and this is an injective (Please refer to the proof of Theorem 2 in Zhang et al. [72] for details). Thus, the output of the computational

| Query | Retrieval | Query | Retrieval |
|-------|-----------|-------|-----------|

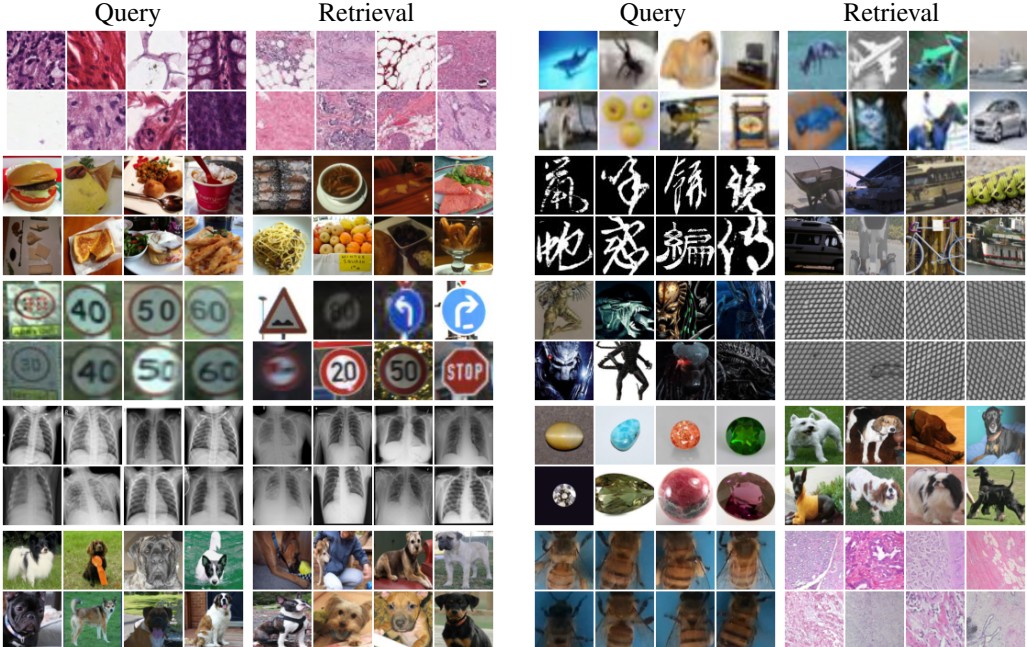

Figure 9: **Retrieved Examples from 10 Meta-test Real-world Datasets.** We present all query-retrieval pairs on meta-test datasets. Each row includes two pairs of query dataset (left) and the retrieved dataset (right). Please see detailed explanations of the pairs in Section D.3 and Table 5.

graph for the network $M$ with the fixed Gaussian input noise $\boldsymbol{x}$ is uniquely represented with the functional encoder $E_{M_F} : M \mapsto \boldsymbol{v}_f$, where $\boldsymbol{v}_f = C_n$ with $n$ nodes on the graph.

Note that we use a network $M$ that is task-adaptively trained for a specific target dataset to not only obtain high performance on the target dataset but also reduce the fine-tuning cost on it. Thus, while we might further need to consider the parameters on the computational graph, we show the injectiveness on the functional encoding only with the computational graph structure and leave the consideration of parameters as a future work, since it is complicated to formally define the injectiveness with trainable parameters.

To sum up, we show the injectiveness of the model representation with both topological encoding and functional encoding schemes, although only one encoding function can injectively represent the entire neural network. While we further concatenate and transform two output representations with a function $g$, to obtain the final model representation: $\boldsymbol{m} = g([\boldsymbol{v}_t, \boldsymbol{v}_f])$, the representation $\boldsymbol{m}$ is also unique on each neural network $M$ with an injective function $g$.

$\square$

Similar to the universal approximation theorem [25, 24], we might construct an injective mapping function $g$ and $\omega$ with learnable parameters on it.

# D   Experimental Setup

## D.1   Architecture Space

Before constructing a model zoo that contains a large number of dataset-architecture pairs, we first need to define an architecture search space on it to handle all architectures in a consistent manner. To easily obtain the task-adaptive parameters for the given task with consideration of various factors, such as a number of layers, kernel sizes, and width expansion ratios, we use the supernet-based OFA architecture space [8], which the same as the well-known MobileNetV3 space [26]. Each neural architecture in the search space consists of a stack of 20 mobile-block convs (MBconvs), where the number of units is 5, and the number of layers on each unit ranges across $\{2, 3, 4\}$. Moreover, for each layer, we select the kernel size is from $\{3, 5, 7\}$, and the width-depth ratio from $\{3, 4, 6\}$. This strategy allows us to generate around $10^{19}$ neural architecture candidates in theory.

## D.2   Model Zoo

To construct a model zoo consisting of a large number of dataset-architecture pairs, we collect 89 real-world datasets for image classification from Kaggle[*] and obtain 100 random architectures per dataset from the OFA space. Specifically, we first divide the collected datasets into two non-overlapping sets for meta-training and meta-testing. If the dataset has more than 20 classes, then we randomly split it into multiple datasets such that a dataset can consist of up to 20 classes. For meta-testing, we randomly selected only one of the splits for each original dataset for diversity. This process yields 140 datasets for meta-training and 10 datasets for meta-testing. To generate a validation set for each dataset, we randomly sample 20% of data instances from each dataset and use the sampled instances for the validation, while using the remaining 80% as the training instances.

For statistics, the number of classes ranges from 2 to 20 with a median of 16, and the number of instances for each dataset ranges from 8 to 158K with a mean of 2,847. We then construct the model-zoo by fine-tuning 100 random OFA architectures on training instances of each dataset and obtaining their performances on its respective validation instances, which yields 14K (dataset, architecture, accuracy) tuples in total. We use this database throughout this paper.

## D.3   Dataset Details

In Table 5, we provide information of all datasets that we utilize for model-zoo construction and meta-test experiments, including the dataset name, a brief description of the dataset, the number of splits for train and test sets, and corresponding Kaggle URL. Please refer to the table if you look into a certain dataset more closely. Particularly, we further provide an explanation of all query and retrieval pairs in Figure 9. Beginning from the left column on the first row, we present pairs of Colorectal Histology (query) & Breast Histopathology (retrieval) and Real or Drawing (query) & Tiny Images (retrieval). In the second row, we show pairs of Dessert (query) & Food Kinds (retrieval) and Chinese Characters (query) & Vehicles (retrieval). For the third row, there are pairs of Speed Limit Signs (query) & Traffic Signs (retrieval) and Alien vs Predator (query) & Grid Anomaly (retrieval). In the fourth row, we illustrate pairs of COVID-19 (query) & Chest X-Rays (retrieval) and Gemstones (query) & Stanford Dogs (retrieval). In the last row, we present pairs of Dog Breeds (query) & Stanford Dogs (retrieval) and Honeybee Pollen (query) & Breast Cancer Tissues (retrieval). Please see Table 5 for the detailed information for each dataset.

## D.4   Baseline NAS Methods

Here we describe the baselines we use in the experiments in the main document. We compare the performance of the models retrieved with our method against pretrained neural networks as well as the ones searched by several efficient NAS methods that are closely related to ours:

**1) MobileNetV3** [26] MobileNetV3 is a representative resource-efficient neural architecture tuned considering mobile phone environments. In our experiments, MobileNetV3 is pretrained on ImageNet-1K, which is fine-tuned for 50 epochs on each meta-testing task.

**2) PC-DARTS** [65], a differentiable NAS method based on a weight sharing scheme that reduces search time efficiently and especially improves memory usage, search time, performance compared to DARTS [35] by designing partial channel sampling and edge normalization. We search for architectures for each meta-testing task by following the official code at https://github.com/yuhuixu1993/PC-DARTS.

**3) DrNAS** [10], a differentiable NAS method that handles NAS as a distribution problem, modeled by Dirichlet distribution. We use the official code at https://github.com/xiangning-chen/DrNAS.

**4) OFA** [8], a NAS method that provides a subnet sampled from a larger network (supernet) pretrained on ImageNet-1K, which alleviates the performance degeneration of prior supernet-based methods. We use the code at https://github.com/mit-han-lab/once-for-all.

**5) MetaD2A** [31], a meta-NAS model that rapidly generates data-dependent architecture for a given task that is meta-learned on subsets of ImageNet-1K. From the ImageNet-1K dataset and architectures of OFA search space, we randomly use 3296 and 14,000 meta-training tasks for the generator and predictor, respectively as a source database.

**6) FBNet** [61], a collection of convolutional models obtained via Differentiable Neural Architecture Search. We use FBNet-A pretrained on ImageNet-1K and fine-tune it on each meta-testing task for 50 epochs.

We use the same hyper-parameters for all baselines for a fair comparison. We fine-tune the architecture for 50 epochs on each meta-testing task. The SGD optimizer is used with a learning rate of 0.01, the momentum of 0.9, and 4e-5 weight decay. The image size is 224×224 and the batch size is 32.

---

[*]https://www.kaggle.com/

Table 3: **Performance Comparison on 5 Unseen Real-world Datasets** All reported results are average performances over 3 different runs with 95% confidence intervals.

| Target Dataset | Method | Params (M) | Search Time (GPU sec) | Training Time (GPU sec) | Speed Up | Accuracy (%) |
|---|---|---|---|---|---|---|
| | MobileNetV3 [26] | 4.00 | - | $178.45_{\pm06.18}$ | $1.00\times$ | $96.86_{\pm0.47}$ |
| | PC-DARTS [65] - 500 Epochs | **3.45** | $943.17_{\pm15.44}$ | $4255.74_{\pm1366.92}$ | $0.03\times$ | $80.48_{\pm14.33}$ |
| | DrNAS [10] - 500 Epochs | 4.12 | $873.44_{\pm25.78}$ | $2445.22_{\pm76.54}$ | $0.05\times$ | $83.58_{\pm2.79}$ |
| Averaged | FBNet-A [61] | 4.30 | - | $218.40_{\pm42.79}$ | $0.82\times$ | $96.15_{\pm2.51}$ |
| Performance | OFA [8] | 6.74 | $121.90_{\pm0.00}$ | $162.33_{\pm03.35}$ | $0.63\times$ | $96.04_{\pm1.00}$ |
| | MetaD2A [31] | 6.15 | $2.56_{\pm0.15}$ | $228.87_{\pm26.64}$ | $0.77\times$ | $97.34_{\pm1.10}$ |
| | **TANS (Ours)** w/ OFA-Based Model-Zoo | 5.50 | $0.002_{\pm0.00}$ | $121.18_{\pm10.71}$ | $1.47\times$ | $97.79_{\pm0.28}$ |
| | **TANS (Ours)** w/ Real-world Model-Zoo | 5.43 | $\mathbf{0.001_{\pm0.00}}$ | $\mathbf{115.06_{\pm16.82}}$ | $\mathbf{1.55\times}$ | $\mathbf{98.59_{\pm0.38}}$ |

# E    Additional Experiments & Analysis

## E.1    Experiment on Real-world Networks

While the proposed TANS shows outstanding performances in a number of neural network search tasks with the manicured architecture search space described in Section D.1, it could be more beneficial if we search the best-fitted model on a given query dataset from the pretrained networks with real-world neural architectures (e.g. ResNet) trained on various datasets. For this even more realistic scenario for Neural Network Search (NNS), we construct our model-zoo including **ten real-world architectures**, such as ResNet18 [23], ShuffleNet V2 [39], MobileNet v2 [45], SqueezeNet [28], GoogLeNet [50], ResNeXt [62], AlexNet [29], MNASNet [53], EfficientNet-B0 [51], and LambdaResNet [4].

**Experimental Setup**    To construct the new **real-world model-zoo**, we first meta-train the real-world architectures and merge the new dataset-network pairs (1,400 pairs) with the random subset of the previous model-zoo (about 5,000 OFA-based models), yielding about 6,500 models in the new model-zoo. The way of training is the same as the experiment introduced in the main document (Section 4.1) except that we only use functional embeddings, while topology information is not used when learning the cross-modal latent space (we exclude the topology information since encoding the topologies of real networks across various search spaces into a single uniform format is too complicated.) Including the real-world architectures, we first verify the retrieval performance of our model on the meta-train datasets, and our TANS achieves 90 for the R@1, 100 for the R@5 scores. The way of evaluating on the meta-test dataset is also the same as the experiments that we conducted in the main document (10 real-world meta-test datasets), except that we conduct experiments only on five datasets out of the ten datasets used in the experiments of the main document, due to the heavy training costs required for meta-testing. The selected datasets are Speed Limit Signs, Honey-bee Pollen, Alien-vs-Predator, Chinese Characters, and COVID-19 datasets (for detailed information for each dataset, please see Table D.3.)

**Experimental Results**    In Table 3, our methods, both with OFA and the real-world architectures, outperform all baseline models, including MobileNetV3 (about $1.0\%p$ to $1.5\%p$ higher), PC-DARTS (about $17.5\%p$ to $18.0\%p$ higher), DrNAS (about $14.5\%p$ to $15.0\%p$ higher), FBNet (about $1.5\%p$ to $2.0\%p$ higher), OFA (about $1.5\%p$ to $2.0\%p$ higher), and MetaD2A (about $0.5\%p$ to $1.0\%p$ higher). We observe that collecting more lightweight real-world neural network and dataset pairs (TANS w/ Real-world Model-Zoo) will allow our model to retrieve computationally efficient pretrained networks in a task-adaptive manner. Such data-driven nature is another advantage of our method since we can easily increase the performance of the model by collecting more pretrained networks that are readily available in many public databases.

## E.2    Additional Performance Comparison with NAS Methods

In the experiment introduced in the main document (Table 1), we train DrNAS and PC-DARTS, which only generate architectures without pretrained weights, for 10 times more iterations (500 epochs) for a fair comparison (while the other methods, which share ImageNet pretrained knowledge, are trained for 50 epochs). In this experiment, rather than training for 500 epochs, we pretrain networks obtained by DrNAS and PC-DARTS on **"ImageNet"** and then fine-tune on two meta-test datasets (Colorectal Histology & Food Classifica-

Table 4: Comparison with NAS methods

| Method | Meta-test Datasets | |
|---|---|---|
| | Colorectal | Food |
| MetaD2A | 96.57% | 89.72% |
| DrNAS w/ ImageNet | 84.27% | 75.90% |
| PC-DARTS w/ ImageNet | 96.77% | 86.75% |
| **TANS 1/10 (Ours)** | 96.83% | **94.31%** |
| **TANS (Ours)** | **97.67%** | 93.71% |

tion Datasets). As shown in Table 4, although pretraining on ImageNet improves their results, our methods, including TANS with 1/10 sized model-zoo (1400), still outperforms all baselines, which shows that retrieving and utilizing **pretrained weights of relevant tasks** is more effective than using ImageNet pre-trained weights.

### E.3 Synergistic Effect of TANS and State-of-the-Art NAS Methods

Not only the real-world architectures but also any existing NAS methods can be successfully integrated with our retrieval framework by simply adding searched networks into our model-zoo. We demonstrate such synergistic effect of TANS and **NAS methods** in Figure 6 (e) of the main document. Constructing the model-zoo with neural architectures generated by MetaD2A, which is a state-of-the-art NAS method, improves our performance compared to the previous model-zoo that are simply sampled from the OFA search space. Considering that NAS approaches have been actively studied [31, 52, 7, 51, 49, 15] and pretrained models are often shared via open-source, we believe that the TANS framework has powerful potential to continuously improve its performance by absorbing such new models into the model-zoo.

## F  Discussion

### F.1  Societal Impacts

Our framework, TANS, has the following beneficial societal impacts: (1) enhanced accessibility, (2) preservation of data privacy, and (3) the reduction of reproducing efforts.

**Enhanced accessibility**   Since our Task-Adaptive Neural Network Search (TANS) framework allows *anyone* to *instantly retrieve* a full neural network that works well on the given task, by providing only a *small* set of data samples, it can greatly enhance the accessibility of AI to users with little knowledge and backgrounds. Moreover, it does not require large computational resources, unlike existing NAS or AutoML frameworks, which further helps with its accessibility. Finally, to allow everyone to benefit from our task-adaptive neural network search framework, we will publicly release our model-zoo, which currently contains *more than 15K models*, and open-source it. Then, anyone will be able to freely retrieve/update any models from our model-zoo.

**Preservation of data-privacy**   Our framework requires only a small set of sampled data instances to retrieve the task-adaptive neural network, unlike existing NAS/AutoML methods that require a large number of data instances to search optimal architectures for the target datasets. Thus, the data privacy is largely improved, and we can further allow the set encoding to take place on the client-side, rather than at the server. This will result in enhanced data privacy, as none of the raw data samples need to be submitted to the system.

**Reduction of reproducing efforts**   Many ML researchers and engineers are wasting their time and labors, as well as the computational and monetary resources in reproducing existing models and fine-tuning them. TANS, since it instantly retrieves a task-relevant model from a model zoo that contains a large number of state-of-the-art networks pretrained on diverse real-world datasets, the users need not redesign networks or retrain them at excessive costs. Since we plan to populate the model zoo with more pretrained networks, the coverage of the dataset and architectures will become even broader as time goes on. Since training deep learning models often requires extremely large computing cost, which is costly in terms of energy consumption, and results in high carbon emissions, our method is also environment-friendly.

### F.2  Limitations

As a prerequisite condition, our method must have a model-zoo which contains pretrained models that can cover diverse tasks and perform well on each given task. There exists a chance that TANS could be affected by biased initialization if the meta-training pool contains biased pretrained models. To prevent this issue, we can use existing techniques that ensure fairness when constructing a model-zoo, which identify and discard inappropriate datasets or models. There have been various studies for alleviating unjustified bias in machine learning systems. Fairness can be classified into individual fairness, treating similar users similarly [16, 68], and group fairness, measuring the statistical parity between subgroups, such as race or gender [70, 36, 22]. Optimizing fair metrics during training is achieved by regularizing the covariance between sensitive attributes and model predictions [59] and minimizing an adversarial ability to estimate sensitive attributes from model predictions [71]. At evaluation times, [3, 12] improves the generalizability for a fair classifier via two-player games. All these methods can be adopted when building our model-zoo.

Table 5: **Dataset Details** Detailed information, such as dataset name, description, and download link, about all datasets that we utilize are described (Due to the space limit, we provide hyperlinks to the webpage for the datasets, rather than printing the full website links.)

| No. | Dataset Name | Brief Description | Instances | Cls. | Splits | URL |
|---|---|---|---|---|---|---|
| MetaTrain-1 | Store Items | Classify store item images by color | 4984 / 624 | 12 | 1 | Link |
| MetaTrain-2 | Big Cats | Classify big cats by species | 2875 / 360 | 4 | 1 | Link |
| MetaTrain-3 | Deepfake Detection | Deepfake detection | 12000 / 1500 | 2 | 1 | Link |
| MetaTrain-4 | Food Kinds | Classify kinds of food | 10580 / 1322 | 11 | 1 | Link |
| MetaTrain-5 | Hair Color | Classify people by hair color | 2560 / 320 | 4 | 1 | Link |
| MetaTrain-6 | Apparels | Classify apparel images by kind and color | 9091 / 1137 | 24 | 2 | Link |
| MetaTrain-7 | Manual Alphabet | Classify manual alphabet letters | 69600 / 8700 | 29 | 2 | Link |
| MetaTrain-8 | Artworks | Classify artworks by artist | 6997 / 877 | 51 | 3 | Link |
| MetaTrain-9 | Blood Cells | Identify blood cell types | 9954 / 1244 | 4 | 1 | Link |
| MetaTrain-10 | Breast Cancer Tissues | Idenify breast cancer with micro-scope images | 6323 / 789 | 8 | 1 | Link |
| MetaTrain-11 | Breast Histopathology | Identify breast cancer with sample images | 222018 / 27753 | 2 | 1 | Link |
| MetaTrain-12 | Aerial Cactus | Identify cacti in aerial photos | 17199 / 2150 | 2 | 1 | Link |
| MetaTrain-13 | Car Models | Classify cars by model | 3229 / 405 | 45 | 3 | Link |
| MetaTrain-14 | Cassava Leaf Disease | Identify type of leaf disease | 17115 / 2141 | 5 | 1 | Link |
| MetaTrain-15 | Celebrity Images | Classify celebrity images by at-tractiveness | 161985 / 20248 | 2 | 1 | Link |
| MetaTrain-16 | Chess Pieces | Identify chess pieces | 437 / 54 | 6 | 1 | Link |
| MetaTrain-17 | Russian Handwrit-ten Letters | Classify Russian handwritten let-ters | 11350 / 1419 | 33 | 2 | Link |
| MetaTrain-18 | CT Images | Identify intracranial hemorrhage in CT scans | 4255 / 532 | 2 | 1 | Link |
| MetaTrain-19 | Corals | Identify types of coral | 489 / 62 | 14 | 1 | Link |
| MetaTrain-20 | Cracks | Detect cracks in pavements and walls | 13570 / 1697 | 2 | 1 | Link |
| MetaTrain-21 | Cactus Identifica-tion | Identify cactus in images | 17199 / 2150 | 2 | 1 | Link |
| MetaTrain-22 | Animals | Classify animal pictures by species | 320 / 32 | 16 | 1 | Link |
| MetaTrain-23 | Blink | Identify which eye is closed | 3874 / 484 | 5 | 1 | Link |
| MetaTrain-24 | Dogs | Classify breeds of dogs | 12558 / 1571 | 120 | 6 | Link |
| MetaTrain-25 | Furniture | Identify type of furniture | 5186 / 648 | 5 | 1 | Link |
| MetaTrain-26 | Forest Fire | Detect whether there is a fire in forest images | 794 / 100 | 3 | 1 | Link |
| MetaTrain-27 | Devanagari Charac-ters | Identify Devanagari characters | 73580 / 9197 | 46 | 3 | Link |
| MetaTrain-28 | COVID Chest X-Ray | Identify COVID by chest x-ray pictures | 599 / 75 | 2 | 1 | Link |
| MetaTrain-29 | Bottles | Identify how full a soda bottle is | 11992 / 1499 | 5 | 1 | Link |
| MetaTrain-30 | Indoor Scenes | Identify the kind of indoor place | 2498 / 312 | 10 | 1 | Link |
| MetaTrain-31 | Flowers | Recognize flower types | 3455 / 431 | 5 | 1 | Link |
| MetaTrain-32 | Four Shapes | Identify basic shapes | 11976 / 1496 | 4 | 1 | Link |
| MetaTrain-33 | Fruits | Identify fruits in different lighting conditions | 35091 / 4386 | 15 | 1 | Link |

*Continued on next page*

| No. | Dataset Name | Brief Description | Instances | Cls. | Splits | URL |
|---|---|---|---|---|---|---|
| MetaTrain-34 | Fruits 360 | Identify fruits in various orientations | 72225 / 9057 | 131 | 7 | Link |
| MetaTrain-35 | Garbage | Classify garbage types | 2019 / 252 | 6 | 1 | Link |
| MetaTrain-36 | Handwritten digits | Identify handwritten digits | 47995 / 5999 | 10 | 1 | Link |
| MetaTrain-37 | Emojis | Identify the type of emojis from various styles | 5324 / 667 | 50 | 3 | Link |
| MetaTrain-38 | German Traffic Signs | Classify german traffic signs | 31367 / 3921 | 43 | 3 | Link |
| MetaTrain-39 | Flowers 2 | Identify type of flower | 5194 / 651 | 102 | 6 | Link |
| MetaTrain-40 | Scraped Images | Classify web images into four generic categories | 27258 / 3408 | 4 | 1 | Link |
| MetaTrain-41 | Natural Images | Classify natural images into six generic categories | 13620 / 1703 | 6 | 1 | Link |
| MetaTrain-42 | Animals and Objects | Identify images of objects and animals | 4000 / 500 | 10 | 1 | Link |
| MetaTrain-43 | Ships | Identify types of ships | 3998 / 500 | 5 | 1 | Link |
| MetaTrain-44 | Surgical Tools | Classify surgical tools | 648 / 81 | 4 | 1 | Link |
| MetaTrain-45 | Land Use | Detect kind of land use from satellite images | 14399 / 1801 | 10 | 1 | Link |
| MetaTrain-46 | Lego Bricks | Classify Lego bricks by shape | 5103 / 638 | 16 | 1 | Link |
| MetaTrain-47 | Lego Bricks 2 | Classify Lego bricks by shape | 7319 / 915 | 20 | 1 | Link |
| MetaTrain-48 | Lego Minifigures | Classify Lego minifigures by franchise | 128 / 14 | 14 | 1 | Link |
| MetaTrain-49 | Real or Fake Legos | Identify off-brand lego bricks from real ones | 36606 / 4576 | 4 | 1 | Link |
| MetaTrain-50 | Makeup | Identify whether a person is wearing makeup | 1203 / 150 | 2 | 1 | Link |
| MetaTrain-51 | Male Female | Identify gender of a person | 46913 / 5864 | 2 | 1 | Link |
| MetaTrain-52 | Messy vs Clean Room | Classify pictures of rooms as either messy or clean | 168 / 22 | 2 | 1 | Link |
| MetaTrain-53 | Cats vs Dogs | Identify cats from dogs | 19975 / 2497 | 2 | 1 | Link |
| MetaTrain-54 | Flowers 3 | Identify type of flower | 3109 / 388 | 5 | 1 | Link |
| MetaTrain-55 | Tiny Images | Identify type of object from tiny images | 15995 / 2000 | 10 | 1 | Link |
| MetaTrain-56 | Mushroom classification | Classify mushrooms by genus | 5312 / 662 | 9 | 1 | Link |
| MetaTrain-57 | Carpet Anomaly | Identify type of anomaly on carpets | 315 / 41 | 6 | 1 | Link |
| MetaTrain-58 | Grid Anomaly | Identify type of anomaly on a grid | 270 / 33 | 6 | 1 | Link |
| MetaTrain-59 | Leather Anomaly | Identify type of anomaly on leather | 293 / 38 | 6 | 1 | Link |
| MetaTrain-60 | Natural Images 2 | Identify types of natural images | 11221 / 1402 | 6 | 1 | Link |
| MetaTrain-61 | Printed Letters | Identify printed Latin letters in various fonts | 381052 / 47630 | 10 | 1 | Link |
| MetaTrain-62 | Bengali Digits | Identify handwritten Bengali digits | 57620 / 7203 | 10 | 1 | Link |
| MetaTrain-63 | Flowers 4 | Identify type of flower | 2855 / 356 | 5 | 1 | Link |
| MetaTrain-64 | Oregon Wildlife | Identify type of wildlife in pictures taken in Oregon | 5655 / 708 | 20 | 1 | Link |
| MetaTrain-65 | Parkinsons Drawings | Identify stage of Parkinson's disease by drawing | 162 / 20 | 2 | 1 | Link |
| MetaTrain-66 | Dogs 2 | Classify types of dogs | 7216 / 902 | 10 | 1 | Link |

| No. | Dataset Name | Brief Description | Instances | Cls. | Splits | URL |
|---|---|---|---|---|---|---|
| MetaTrain-67 | Seedlings | Determine type of plant from a picture of its seedling | 3792 / 474 | 12 | 1 | Link |
| MetaTrain-68 | Traffic Signs | Identify traffic signs | 5735 / 717 | 8 | 1 | Link |
| MetaTrain-69 | Real vs Fake Faces | Identify fake face images from real ones | 1632 / 204 | 2 | 1 | Link |
| MetaTrain-70 | Casting Products | Idendify defects in products manufactured by casting | 6866 / 858 | 2 | 1 | Link |
| MetaTrain-71 | Rock Paper Scissors | Identify hand gestures | 1749 / 219 | 3 | 1 | Link |
| MetaTrain-72 | Chest X-ray | Identify various information from chest x-ray images | 4484 / 560 | 2 | 1 | Link |
| MetaTrain-73 | Furniture 2 | Classify type of furniture | 2400 / 200 | 200 | 10 | Link |
| MetaTrain-74 | Sheep | Classify sheep breeds | 1344 / 168 | 4 | 1 | Link |
| MetaTrain-75 | Simpsons | Identify characters from a popular TV show | 15969 / 1999 | 39 | 2 | Link |
| MetaTrain-76 | Simpsons 2 | Identify characters from a popular TV show | 16709 / 2090 | 39 | 2 | Link |
| MetaTrain-77 | Skin Cancer | Classify type of Skin Cancer | 1785 / 224 | 9 | 1 | Link |
| MetaTrain-78 | Signed Digits | Identify sign language digits | 1644 / 208 | 10 | 1 | Link |
| MetaTrain-79 | Stanford Dogs | Identify dog breeds | 16376 / 2050 | 120 | 6 | Link |
| MetaTrain-80 | Preprocessed Stanford Dogs | Preprocessed version of Stanford Dogs | 16418 / 2052 | 120 | 6 | Link |
| MetaTrain-81 | Synthetic Digits | Identify digits on randomly generated backgrounds | 9600 / 1200 | 10 | 1 | Link |
| MetaTrain-82 | Ethiopic Digits | Classify Ethiopic Digits | 48000 / 6000 | 10 | 1 | Link |
| MetaTrain-83 | Simpsons 3 | Identify characters from a TV show | 16709 / 2090 | 39 | 2 | Link |
| MetaTrain-84 | Traffic Signs 2 | Identify traffic signs | 20288 / 2530 | 67 | 4 | Link |
| MetaTrain-85 | Vehicles | Classify types of auto vehicles | 22427 / 2803 | 17 | 1 | Link |
| MetaTrain-86 | Clothes | Identify types of clothes | 12935 / 1617 | 6 | 1 | Link |
| MetaTest-1 | Alien vs Predator | Tell apart characters from a movie | 711 / 89 | 2 | - | Link |
| MetaTest-2 | Colorectal Histology | Classify colorectal tissue images | 4000 / 496 | 8 | - | Link |
| MetaTest-3 | COVID-19 | Identify lung diseases from radiographic images | 2298 / 288 | 3 | - | Link |
| MetaTest-4 | Speed Limit Signs | Classify road speed limit signs | 272 / 35 | 4 | - | Link |
| MetaTest-5 | Gemstones | Classify different kinds of gemstones | 2206 / 278 | 18 (87)$^\dagger$ | - | Link |
| MetaTest-6 | Honeybee Pollen | Detect whether a honeybee is carrying pollen | 571 / 71 | 2 | - | Link |
| MetaTest-7 | Chinese Characters | Identify handwritten Chinese characters | 13762 / 1715 | 20 (200)$^\dagger$ | - | Link |
| MetaTest-8 | Real or Drawing | Identify real images from drawings in tiny images | 4000 / 500 | 10 | - | Link |
| MetaTest-9 | Dessert | Identify types of dessert | 1324 / 166 | 5 | - | Link |
| MetaTest-10 | Dog Breeds | Classify dog breeds | 5295 / 656 | 19 (133)$^\dagger$ | - | Link |

$^\dagger$The original dataset's number of classes are written in parentheses.