# OpenReview forum: "Task-Adaptive Neural Network Search with Meta-Contrastive Learning"
_NeurIPS.cc/2021/Conference — NeurIPS 2021 Spotlight_

### Official Review · Reviewer_ZAGc · 2021-07-16

**Rating:** 6
**Confidence:** 4

**Summary:**

This paper introduces an interesting problem of Neural Network Search (NNS), aiming at searching for the optimal pretrained network for a novel dataset from a model zoo. The authors propose a framework TANS to tackle this problem. The proposed method constructs a model encoder and a dataset encoder to calculate the embedding of model and query set respectively. By applying the contrastive loss to maximize the similarity between a dataset and a network that obtains high performance on it, the framework is trained jointly with a surrogate performance predictor. At testing time, the framework is able to retrieve the well-fit architectures from the model zoo given an unseen dataset.

The main contribution of this paper is that it introduces a novel problem Neural Network Search and also present a decent method to tackle the problem.

**Limitations And Societal Impact:**

The author has provided in the supp.

**Main Review:**

Pros:
1. The paper is well written and easy to follow.
2. The introduced problem NNS is novel to me. And the problem is practical and does exist in the real deployment scenario.
3. The proposed method is technically sound and the authors also have provided sufficient details and ablation study to validate the effectiveness.

Cons:
1. I'm confused about the proposed functional embeddings. I'm not sure why introducing it since we've already have a topology embeddings. Does it play a significant role in the proposed framework. In table 2, the author has conducted an ablation w.r.t. the topol embedding but fail to conduct a similar one on functional embedding. I would like to see this ablation.

2. I would like to see an ablation study on the surrogate model. Does the predicted performance correlate with the GT acc well?

3. In Fig. 6c, the authors has visualized the the cross-model latent space to demonstrate the claim that "the network whose embeddings are the closest to the query dataset achieves higher performance on it" (L358). However, I think the best way to validate this point is to calculate the correlation between the latent distance and the real performance to see if they correlate well.

4. The authors also show the effect of model-zoo size on accuracy in Fig. 6d. I'm wondering the concrete data reduction method here? Do the author simply shrink the data size of each dataset or just simply remove some datasets? These two methods might have different impact on the performance. I would like the authors to clarify this and it would be better to see the results of both two methods.


**Time Spent Reviewing:**

3.0

---

> ### Author Response · Authors · 2021-08-09
> **Response to reviewer ZAGc**
>
> **(1) I'm confused about the proposed functional embeddings. I'm not sure why introducing it since we've already had topology embeddings. Does it play a significant role in the proposed framework? In table 2, the author has conducted an ablation w.r.t. the Topol embedding but fails to conduct a similar one on functional embedding. I would like to see this ablation**
>
> - Please note that the topology embedding can only represent the **architecture** of the neural network, but not the parameters. This means that with the topology embedding only, we cannot distinguish between two networks with the same architecture trained on different datasets. Thus, TANS which uses the topology embedding only is not a meaningful baseline as it cannot retrieve the model that is trained on a relevant dataset. The table below reports the recall performance of TANS with only the topology embedding (TANS w/o Func Emb.), which is better than random but highly inaccurate. On the other hand, TANS with functional embeddings can retrieve models trained on relevant datasets. Thus both types of embeddings are essential to retrieve the networks with “best-performing architectures” trained on “relevant datasets”, for a given dataset.
> - We have discussed this in Line 305-314, but will further clarify this point.
>
>
> |                     |  R@1  |  R@5  |  R@10 | Mean  | Median |
> |---------------------|:-----:|:-----:|:-----:|-------|--------|
> | Random              |  2.14 |  2.86 |  8.57 | 69.04 |  70.0  |
> | TANS w/o Func. Emb. |  5.00 | 11.43 | 18.57 | 63.20 |  63.0  |
> | **TANS (Ours)**               | **80.71** | **96.43** | **99.29** |  **1.90** |   **1.0**  |
>
> ***
>
> **(2) I would like to see an ablation study on the surrogate model. Does the predicted performance correlate with the GT acc well?**
>
> - As you suggested, we conduct an ablation study on our performance surrogate model. However, please note that **we do not use ranking loss, but a contrastive loss**, which does not rank the negative examples. Thus, while we can expect models trained on semantically relevant datasets, that perform well on the query dataset, to be close to it, **we cannot expect the negative examples to be well-ranked**. That is, the negative examples may be in any order possible with the contrastive loss. Thus we use **Mean Squared Error (MSE)** scores to show how accurately our performance predictor estimates the accuracy given dataset, architecture, and parameters embeddings rather than Spearman's or Pearson's correlation scores.
>
> - First, we retrieve the top 10 most relevant models for an unseen query datasets (we pick 5 datasets among 10 datasets for computational convenience) and then compute the MSE between the estimated scores predicted by our performance surrogate model and the actual ground truth accuracies of the models. For an ablation study, we eliminate the query and the model embeddings from its input embeddings, respectively. The results are shown in the below table. As shown, we observe that removing either query or model embeddings degrades performance compared to the predictor taking both embeddings. It is natural that, with only model or query information, it is difficult to correctly estimate the accuracy since the predictor fails to recognize what or where to learn. With both embeddings (our proposed predictors), we observe that we achieve significantly improved MSE scores.
>
> - To further validate the accuracy of the performance predictor, we report the MSE between the predicted performance using the predictor and the ground truth performance of each model for the entire set of pretrained models from a smaller model zoo (we use a smaller one since this requires to obtain GT performance on all models). Although the performance predictor achieves slightly higher MSE scores for this experiment compared to the MSE obtained on the top-10 retrieved models (which are more important), the MSE scores are still meaningfully low, which implies that our performance surrogate model accurately estimates the performance of the models from the model zoo.
>
> | MSE on Top 10 Retrieved Models   |  Food  | Gemstones | Dog Breeds | Alien v.s. Predator | COVID-19 |
> |--------------------------|:------:|:---------:|:----------:|---------------------|----------|
> | Random  | 0.1454    |    0.1175 |     0.1622 |         0.2631 |       0.1881 |
> | Predictor w/o Query Embedding  | 0.0178    |    0.0782 |     0.0194 |         0.0185 |       0.0418 |
> | Predictor w/o Model Embedding  |  0.0188   |  **0.0323**   |      0.0016 |       0.0652  |      0.03279   |
> | Predictor (Ours)                           | **0.0036**     |    0.0338 |     **0.0013** |              **0.0028** |   **0.0233** |
>
> | MSE on our Model-Zoo      |  Food  | Gemstones | Dog Breeds | Alien v.s. Predator | COVID-19 |
> |--------------------------|:------:|:---------:|:----------:|---------------------|----------|
> | Random            | 0.1619 |    0.1081 |     0.1348    |    0.2609          |    0.2928     |
> | Top 10 Retrieved Models             | 0.0036 |    0.0338 |     0.0013 |              0.0028 |   0.0233 |
> | 1/100 Sized Model-Zoo               | 0.0088 |    0.0369 |     0.0034 |              0.0077 |   0.0241 |
>
> ***
>
> **(3) In Fig. 6c, the authors have visualized the cross-modal latent space to demonstrate the claim that "the network whose embeddings are the closest to the query dataset achieves higher performance on it" (L358). However, I think the best way to validate this point is to calculate the correlation between the latent distance and the real performance to see if they correlate well.**
>
> - Thank you for your suggestion. Please note that measuring correlation with the distances is not compatible with the contrastive loss we use, since the negative examples (models that achieve low performance on the target dataset) are pushed away from the query, without a meaningful ranking between the negative instances. Thus $999_{th}$-best model could have lower performance than a $1,200_{th}$-best model. The reason we use contrastive loss, is because we only want a few models that work well for the given query dataset, and do not need to retrieve models that do not work well on it, unlike image or document retrieval which aims to retrieve a large list of items. To obtain a latent space where the negative examples are also well-ranked, we should replace the contrastive loss with a ranking loss instead, but this will not be very meaningful.
>
> - However, as recommended, we examined the correlation between the distance of the model (similarity) and the actual performance on the query dataset. Since our latent space does not learn meaningful rankings between negative examples as explained, but rather learns clusters, we selected 50 models around the query point, and selected 50 models around the farthest points, thus using a total of 100 models to report the correlation scores. In the table below, we show the correlation scores of these 100 models on the five unseen datasets. For Food dataset (reported as **"hard"** dataset in Table 1 of the main paper), the correlation scores are shown to be high. On the other hand, for Colorectal Histology dataset (reported as **"easy"** dataset), the correlation scores are low as any model can obtain good performance on it, which makes the performance gap across models small. In sum, as the task (dataset) becomes more difficult, we can observe a higher correlation in the latent space between the distance and the rank. Thank you for the insightful suggestion, and we will include this result in the revised version of the paper.
>
> |                   | Food  | Drawing |Chinese Char. | Alien vs. Predators | Colorectal Histology |
> |-------------------|-------|---------------|---------|---------------------|---------------|
> | Spearman's Corr. | 0.752 | 0.583         | 0.322   | 0.214               | 0.213         |
>
> ***
> **(4) The authors also show the effect of model-zoo size on accuracy in Fig. 6d. I'm wondering about the concrete data reduction method here? Do the author simply shrink the data size of each dataset or just simply remove some datasets? These two methods might have a different impact on the performance. I would like the authors to clarify this and it would be better to see the results of both two methods.**
>
> - In the “Random” method, the model zoo is reduced by randomly selecting a subset of the dataset-model pairs from the 14,000-pair model zoo. In the “Ours” (Efficient Construction) method, the algorithm is described in Section A.4 in the supplementary material, and the experimental details are described in Section B.3 (Lines 82-86).

---

### Official Review · Reviewer_6K2a · 2021-07-17

**Rating:** 7
**Confidence:** 4

**Summary:**

This paper proposes and tackles the following task: given an unseen dataset, selecting the optimal pre-trained neural network from a model zoo of networks which will obtain the best fine-tuned performance on the new dataset. To solve this problem they propose a novel contrastive learning method to learn a shared embedding space for the models and datasets which minimizes the distance of the dataset embeddings and model embeddings of models that can be well adapted to the datasets. They propose encoding the properties of a pretrained model using a combination of a topological and functional encoding and demonstrate the effectiveness of this encoding scheme. They also propose a novel method for efficient model-zoo construction and build a dataset and train their algorithm on a collection of Kaggle datasets. They demonstrate that they can achieve very high performance on unseen datasets with few training steps.

**Limitations And Societal Impact:**

They discuss it in the appendix. They mostly address the societal impacts, but maybe could include some additional discussion of the possible impact of biased initializations.

**Main Review:**

This paper tackles a novel problem of building a model zoo of pretrained networks and retrieving the optimal base model for fine tuning on an unseen dataset. This is a valuable direction of research and has strong significance for practical machine learning since picking an existing trained model from a model zoo and fine tuning it for your task is an extremely common method to quickly train an effective model on a new dataset. The methodology also includes a novel method of characterizing the model using a combination of topological and function encoding.

Unfortunately, I currently can't recommend acceptance for this paper. I believe that this work requires a stronger baseline to demonstrate the generalization of the dataset embedding.

Additional results would help ablate the effectiveness of the dataset embedding and the quality of the models and pretrained weights from the model zoo. While table 3 does ablate the choices made for the embedding, it's not entirely clear how this generalizes to unseen datasets from just those experiments. A strong baseline  that would significantly improve the quality of the paper would be for each unseen test dataset X, sample 5-10 random other test datasets, and retrieve and train the optimal models for those datasets on dataset X. This would help ablate the effectiveness of choosing random strong models from the model zoo with the effectiveness of the model embedding. In table 3, the author's do show some ablations of the different retrieval strategies which helps demonstrate their choices for the retrieval method. However, this could be strengthened by making sure if the models selected, but not trained on the specified data still performed adequate on the specified dataset.

The paper would also be much improved with more analysis of the efficiency of the methodology especially related to how much compute is used to generate the Model zoo. This is especially important to demonstrate the efficacy of their novel efficient model zoo construction method.

The results that they demonstrate on the unseen dataset are impressive, but difficult to directly compare. None of the existing models were pre-trained on the same training kaggle datasets and the model zoo was trained on both those and the imagenet dataset through the OFA initialization. It would also be helpful to compare the models if the flops of each model was included in Table 1. Something that help demonstrate their algorithm would be testing their ability to transfer to common unseen datasets like cifar-10/cifar-100/imagenet.

The paper is generally clear and easy to follow, but some of the details could be clearer. How are the positive and negative pairs selected? How exactly is the meta performance surrogate model used? How were models trained to create the model zoo? It seems both a 14,000 model model zoo and an efficiently constructed model zoo were created? It's not entirely clear which is used in which experiments. Was the model zoo trained off OFA in the meta-fashion or each architecture separately?

==================================================================================================

Post-Rebuttal Update:
I would like to thank the authors for addressing many of my concerns and especially including the additional baselines and experiments. I believe that the baseline exploring the capabilities of the models in just the model zoo significantly improves the papers and I think the paper would benefit from the inclusion of the full baseline. In addition, the inclusion of results with training the competitors on the same kaggle meta-training addressed my concerns on fairness with regards to pretraining on the kaggle datasets. The authors have definitely demonstrates the effectiveness of your task-adaptive retrieval mechanism and addressed my main concern. Given that and the inclusion of more discussion on the comparative pretraining costs, and additional experiments demonstrating the robustness of the method on other datasets, I would recommend acceptance for this paper and raise my score to a 7.

**Time Spent Reviewing:**

5

---

> ### Author Response · Authors · 2021-08-08
> **Response to reviewer 6K2a - Part 2**
>
> **(6) How are the positive and negative pairs selected?**
>
> - We did describe how positive and negative pairs are selected for meta-constrastive learning in **Section B.1** **(Line 61-63)** of the supplementary file. Our model-zoo contains 14,000 pre-trained models (140 datasets * 100 architectures). At each training iteration of meta-contrastive learning, we sample a model for each of the 140 unique datasets, and use it as a positive example. Then, for each dataset, we choose a model from other datasets (from the same batch) as a negative example. We will include this in the main paper.
>
> ***
>
> **(7) How exactly is the meta performance surrogate model used?**
>
> - This is clearly described in the main paper, and we also have ablation studies that show the effectiveness of the meta performance predictor. The meta performance surrogate model is used to (1) guide the learning of the cross-modal space **(Line 168-17)**, and (2) to select the best-fitted models from retrieved candidates explained **(Line 192-198)**. The effectiveness of the meta performance predictor in the learning of TANS is shown in Table 3, as the TANS w/o the predictor yields lower recall scores than TANS trained with the performance predictor. In Figure 7, we show that the meta-performance predictor is effective in selecting the best fitted models, by reporting the performance gains using it (See 330-340 for more explanation).
>
> ***
>
> **(8) How were models trained to create the model zoo?**
>
> - This is described in **Section B.3 (Line 77-81)** of the supplementary file. For the model-zoo consisting of 14,000 random pairs used in the main experiment, we fine-tune the ImageNet1K-pretrained OFA models on each of the 140 datasets for 625 epochs, following in the “progressive shrinking” method described in [1]. We then choose 100 random OFA architectures for each dataset and evaluate their test accuracies on the test split.
>
> ***
>
> **(9) It seems both a 14,000 model zoo and an efficiently constructed model zoo were created? It's not entirely clear which is used in which experiments.**
>
> - We used the model-zoo containing 14,000 models as our base model-zoo unless otherwise stated, and we clearly stated which model zoo we use for each result. For example, the meta-test experiment leverages 14,000 model-zoo (Line 262-264), the ablation study for architectures and parameters uses 1/20 sized model-zoo (Line 315-317), and the visualization of the latent space utilize 1/10 sized model-zoo (Line 356-357).
>
> ***
>
> **(10) Was the model zoo trained off OFA in the meta-fashion or each architecture separately?**
>
> - As described in **Section B.3 (Line 77-81) of the supplementary file**, we employed the ‘progressive shrinking’ as described in [1] in order to save computation by training all of the OFA subnets simultaneously, rather than training each architecture one-by-one. The Progressive Shrinking method is as follows. For each meta-train dataset, first, we train the OFA supernet for 125 epochs. Next, we fine-tune the subnets in four stages. One stage consists of 150 epochs, where at each training step, we randomly sample a subnet and train it. At each stage, we allow smaller and smaller subnets to be sampled, and in the last stage, all subnets have an equal probability of being sampled. This way, we effectively train all OFA subnets simultaneously.
>
> ***
>
> **(11) They mostly address the societal impacts, but maybe could include some additional discussion of the possible impact of biased initializations.**
>
> - Thank you for the helpful suggestion. We will include the following discussion in the societal impacts section: There exists a chance that TANS could be affected by biased initialization if the meta-training pool contains biased pretrained models. To prevent this issue, we can use existing techniques that ensure fairness when constructing a model-zoo, which identify and discard inappropriate datasets or models. There have been various studies for alleviating unjustified bias in machine learning systems. Fairness can be classified into individual fairness, treating similar users similarly [2,3], and group fairness, measuring the statistical parity between subgroups, such as race or gender [4,5,6]. Optimizing fair metrics during training is achieved by regularizing the covariance between sensitive attributes and model predictions [7] and minimizing an adversarial ability to estimate sensitive attributes from model predictions [8]. At evaluation times, [9,10] improves the generalizability for a fair classifier via two-player games. All these methods can be adopted when building our model-zoo.
>
> ***
>
> We believe that we have faithfully addressed all your concerns, and politely ask you to revise the score and the review accordingly. We thank you again for your time and efforts in reviewing our paper.
>
> ***
>
> **References**
> - [1] Han Cai, Chuang Gan, Tianzhe Wang, Zhekai Zhang, and Song Han. Once for all: Train one
> network and specialize it for efficient deployment. In International Conference on Learning Representations, 2020.
> - [2] Cynthia Dwork, Moritz Hardt, Toniann Pitassi, Omer Reingold, and Richard Zemel. Fairness
> through awareness. In Proceedings of the 3rd innovations in theoretical computer science conference, pp. 214–226, 2012.
> - [3] Mikhail Yurochkin, Amanda Bower, and Yuekai Sun. Training individually fair ml models with
> sensitive subspace robustness. In International Conference on Learning Representations, 2019.
> - [4] Rich Zemel, Yu Wu, Kevin Swersky, Toni Pitassi, and Cynthia Dwork. Learning fair representations. In International Conference on Machine Learning, pp. 325–333, 2013.
> - [5] Christos Louizos, Kevin Swersky, Yujia Li, Max Welling, and Richard Zemel. The variational fair autoencoder. arXiv preprint arXiv:1511.00830, 2015.
> - [6] Moritz Hardt, Eric Price, and Nati Srebro. Equality of opportunity in supervised learning. In
> Advances in neural information processing systems, pp. 3315–3323, 2016.
> - [7] Blake Woodworth, Suriya Gunasekar, Mesrob I Ohannessian, and Nathan Srebro. Learning nondiscriminatory predictors. In Conference on Learning Theory, pp. 1920–1953, 2017.
> - [8] Brian Hu Zhang, Blake Lemoine, and Margaret Mitchell. Mitigating unwanted biases with adversarial learning. In Proceedings of the 2018 AAAI/ACM Conference on AI, Ethics, and Society, pp. 335–340, 2018a.
> - [9] Alekh Agarwal, Alina Beygelzimer, Miroslav Dudik, John Langford, and Hanna Wallach. A reductions approach to fair classification. In International Conference on Machine Learning, pp.
> 60–69, 2018.
> - [10] Andrew Cotter, Maya Gupta, Heinrich Jiang, Nathan Srebro, Karthik Sridharan, Serena Wang, Blake Woodworth, and Seungil You. Training well-generalizing classifiers for fairness metrics and other data-dependent constraints. In International Conference on Machine Learning, pp. 1397–1405, 2019.

---

> > ### Comment · Reviewer_6K2a · 2021-08-21
> > **Thanks for the clarifications and additional experiments.**
> >
> > I would like to thank the authors for addressing many of my concerns and especially including the additional baselines and experiments. I believe that the baseline significantly improves the papers and I think the paper would benefit from the inclusion of a full baseline. It definitely demonstrates the effectiveness of your task-adaptive retrieval mechanism.
> >
> > I would like to recommend acceptance for this paper if you could help me understand and analyze a final concern I had with the baseline results.
> >
> > I believe some of the results from that baseline may inconclusively contradict some of your claims. While I can only compare the (Hard) Food task with the results in your paper since you don't have all the tasks and the TAN are quite better than the random baseline, the best score out of five networks of 90.12% is better than all your other baselines including the 89.72 MetaD2A baseline which you claim is a fair comparison. This may seem to indicate that the comparison with other methods isn't quite as fair as indicated and your specific pre-training method (possibly because it includes kaggle) isn't quite that fair a comparison with all the other baselines. Obviously results on one task and the reason for this is inconclusive, but I believe that this paper would still strongly benefit from a very careful analysis of this area.
> >
> > The inclusion does seem to show via the ablation the large benefit of your task-adaptive retrieval however. Your inclusion of the results with pretraining the other Baselines on the kaggle datasets also seems to contradict this reasoning. Thanks for the inclusion and I think it helps alleviate some of my concerns. So I would really appreciate if you could help me explain to me how these results don't contradict or include clarifying experiments (maybe more task comparisons will show Food task is an outlier) or analysis. Was Meta2DA trained on your 14k model zoo?
> >
> > The inclusion of the transfer to CIFAR-10/CIFAR-100 helps demonstrate the generalizability and improves the paper.
> >
> > Thanks for clarifying the remaining details of your paper. The the part I'm don't believe is very clear is how the performance surrogate model "guide the learning of the cross-modal space". How does it affect L_m or L_q? Does is it guiding learning just via shared features or parameters or does it affect training explicitly.
> >
> > Thanks for the clarification on the model-zoo details and training times. It may be useful in the final paper to acknowledge the difference in pretraining cost (I believe Meta2DA for example used ~21.1 GPU hours and you are also starting generation with a fully trained OFA network) and include a more direct comparison of your efficient model zoo construction method with the full zoo.

---

> > > ### Author Response · Authors · 2021-08-21
> > > **A quick response**
> > >
> > > I would like to thank the authors for addressing many of my concerns and especially including the additional baselines and experiments. I believe that the baseline significantly improves the papers and I think the paper would benefit from the inclusion of a full baseline. It definitely demonstrates the effectiveness of your task-adaptive retrieval mechanism.
> > >
> > > - We sincerely thank you for your constructive suggestions, and will include the results of this baseline in the revision.
> > >
> > > ---
> > >
> > > I believe some of the results from that baseline may inconclusively contradict some of your claims. While I can only compare the (Hard) Food task with the results in your paper since you don't have all the tasks and the TAN are quite better than the random baseline, the best score out of five networks of 90.12% is better than all your other baselines including the 89.72 MetaD2A baseline which you claim is a fair comparison. So I would really appreciate if you could help me explain to me how these results don't contradict or include clarifying experiments (maybe more task comparisons will show Food task is an outlier) or analysis.
> > >
> > > - First of all, please note that neither the random retrieval baseline or MetaD2A, or any of the existing methods, can correctly select the task-relevant parameters, and thus **MetaD2A has no advantage over the random retrieval baseline in terms of the quality of the parameters selected**. Actually, being able to select task-relevant pretrained network is a unique ability of our method that cannot be achieved by any of the existing works, including MetaD2A, and this is the main novelty and the strength of our framework. Also, achieving such an ability is challenging due to the lack of good network encoder, and the lack of a proper method to learn a latent space where semantically relevant dataset and the pretrained networks are embedded closer to each other, both of which we tackle in this paper.
> > >
> > > - MetaD2A, however, does have the ability to **generate** task(dataset)-dependent architectures. This ability to generate optimal architectures that are not seen in the meta-training pool helps it outperform the random retrieval baseline that can only retrieve from the given pool of pretrained networks, as shown in the table below, although it largely underperforms ours.
> > >
> > > |                  |    Food    |  Gemstones | Traffic Signs | Dog Breeds | Chinese Char. |
> > > |------------------|:----------:|:----------:|:-------------:|------------|---------------|
> > > | Random Retrieval | 89.16±0.88 | 90.09±3.13 |   93.80±1.44  | 95.03±0.59 |   94.89±0.79  |
> > > | MetaD2A          | **89.72±1.35** | **92.72±1.28** |  **94.37±2.44**  | **96.11±1.47** |   **96.87±1.25**  |
> > >
> > > - The Food dataset is not an outlier. Note that the reported performance of MetaD2A baseline is the **average over 3 random runs, and not the best performance**. If you average the performance obtained by the random retrieval baseline you suggested, it is **89.16**, which is lower than **89.72** achieved by MetaD2A. Moreover, the best performance obtained using MetaD2A is **91.07**, which is significantly higher than 90.12 obtained using the random retrieval baseline. Finally, as explained before, MetaD2A has no knowledge of the task-relevant parameters as it only learns to generate architectures.
> > >
> > > - The comparison with MetaD2A is fair, since it is meta-trained on the **identical meta-training set of neural architectures** to one used by TANS for meta-training, and uses a supernet pretrained on a random Kaggle dataset from the meta-training pool, .
> > >
> > > ---
> > >
> > > This may seem to indicate that the comparison with other methods isn't quite as fair as indicated and your specific pre-training method (possibly because it includes kaggle) isn't quite that fair a comparison with all the other baselines. Obviously results on one task and the reason for this is inconclusive, but I believe that this paper would still strongly benefit from a very careful analysis of this area.
> > >
> > > - This is a misunderstanding. For the new results we report in Part 1 of our initial response, all baselines are trained on **one of the randomly selected Kaggle datasets from the same meta-training pool**, and each of the pretrained network in our model zoo is trained on **a single dataset from the Kaggle database**. The only difference is that TANS can retrieve task-relevant pretrained network for the query while baselines cannot. Please check the below Table, which includes the new results (further details can be found in the response to the comment (2) in our initial response).
> > >
> > > |    Method   | Params | MFLOPs | Search Time(sec) | Training Times(sec) | Speed Up | Acc. (%) |
> > > |:-----------:|:------:|:------:|-------------|----------------|----------|----------|
> > > | MobileNetV3 |   4.21     |  132.94       |   -          |   402.48 ± 99.54           |    1.00x      |    90.03 ± 04.01       |
> > > | OFA         |      6.00  |     **148.76**   |       121.90      |          500.35 ± 67.07      |     0.65x     |     93.12 ± 01.58     |
> > > | MetaD2A     |    6.26    |  512.67      |      2.59       |            1228.82 ± 800.38    |     0.32x     |     93.42 ± 01.50     |
> > > | TANS (Ours) |   **5.51** | 181.74 |      **0.002**  |       **200.93 ± 24.21**  |     **1.98x** |   **96.28 ± 00.24** |
> > >
> > > - Please note that there is no **dataset-, class-, and instance-level overlap** across the meta-training and meta-test set, as clearly described and emphasized in Line 260-261 of the main paper. Thus, none of the instances or the classes encountered at the meta-test time are seen during meta-training. Also, the datasets are highly heterogeneous, although they are from the same Kaggle database (Please see Figure 4). Even if a pretrained model is trained on the dataset from the same Kaggle database, if it is highly irrelevant to the query task, its performance can be very low, as shown by the performance of the farthest models in the cross-modal retrieval space, in Figure 6.
> > >
> > > - TANS also outperforms all other baselines even on **CIFAR-10** and **CIFAR-100 datasets** that are largely different from the datasets in the Kaggle database, as shown in additional experimental results we provided in the Part 1 of our response.
> > >
> > > ---
> > >
> > > Was Meta2DA trained on your 14k model zoo?
> > >
> > > - Yes. MetaD2A is meta-trained on the exactly the same set of architectures with the pretrained models in our 14k model zoo. However, note that MetaD2A is a conventional NAS method that can only learn to generate architectures, and **cannot learn anything from the parameters**. Thus the knowledge of the architectures from the 14K model zoo is used for its meta-training, but not the parameters.
> > >
> > > ---
> > >
> > > Thanks for clarifying the remaining details of your paper. The the part I'm don't believe is very clear is how the performance surrogate model "guide the learning of the cross-modal space". How does it affect L_m or L_q? Does is it guiding learning just via shared features or parameters or does it affect training explicitly.
> > >
> > > - As described in Line 171-173, our performance surrogate model $S$ takes the query embedding $q^\tau$ and the model embedding $m^\tau$ as an input for the given task $\tau$. Then, we minimize the mean-squared error loss $(s_{acc}^{\tau} - S(m^\tau,q^\tau))^2$, where $s_{acc}^{\tau}$ is the ground truth accuracy (Line 174-175). When optimizing this performance predictor, gradients updates calculated for its inputs $m^\tau$ and $q^\tau$ are **backpropagated to both the model and query encoders**. Hence, our performance predictor explicitly affects the learning of the cross-modal latent space. We have shown the effectiveness of this explicit guidance given by the performance predictor with an **ablation study**, which reports the performance of TANS w/o Predictor in **Table 3**, which shows that it achieves lower recall performance compared to our full model (TANS+Contrastive).
> > >
> > > ---
> > >
> > > Thanks for the clarification on the model-zoo details and training times. It may be useful in the final paper to acknowledge the difference in pretraining cost (I believe Meta2DA for example used ~21.1 GPU hours and you are also starting generation with a fully trained OFA network) and include a more direct comparison of your efficient model zoo construction method with the full zoo.
> > >
> > > - Thank you for the suggestions. We will include the difference in pretraining cost between the baselines and TANS in the final paper. However, we emphasize that unlike other NAS baselines such as MetaD2A, PC-DARTS, and DrNAS, building the model zoo for TANS is a one-time cost, and once built, TANS can repeatedly reuse the model zoo for any number of tasks (datasets) at zero building cost.
> > > - We will also include a more detailed comparison between the efficient model zoo construction method and the larger model zoo in terms of construction cost and the final performance, in the revision.

---

> > > ### Author Response · Authors · 2021-08-28
> > > **The end of the discussion phase is approaching**
> > >
> > > Dear Reviewer 6K2a,
> > >
> > > We now have less than a week to have interactive discussions. We have faithfully addressed your concern and misunderstanding on the fairness of the comparison, and would like to gently remind you about it. The following is the quick summary of the response.
> > >
> > > - We provided the average performance of MetaD2A and the random retrieval baseline you suggested on five different datasets. The results show that **MetaD2A outperforms this random retrieval baseline on all query datasets**. The best performance obtained by MetaD2A on the Food dataset is also **much higher**than one obtained by the random retrieval baseline.
> > > - **All baselines** in the new experimental results we provided in the initial response, and below, are **trained on the Kaggle dataset**, including MetaD2A.
> > > - Finally, **neither MetaD2A nor any of the baselines has the ability to retrieve the best-performing pretrained network for a given dataset**, and this is a unique ability of our TANS framework. MetaD2A, however, outperforms the random retrieval baseline since it can find optimal architecture for a given dataset.
> > > - The performance predictor **directly affects the learning of the cross-modal retrieval space**, as its gradient is backpropagated to both the dataset and the model encoder.
> > >
> > > Please let us know if there is anything else you want us to clarify. We thank you again for your constructive comments.
> > >
> > > Best regard,
> > > The authors

---

> > > > ### Comment · Reviewer_6K2a · 2021-08-28
> > > > **I agree with fairness of comparisons. Unclear to me that comparing average performance of random search is comparible**
> > > >
> > > > I would like to thank the authors for their discussion with me and providing the additional information. I believe that your experiments definitely demonstrates the advantages of your method full method and mostly demonstrate the fairness of the comparisons. However, I think the performance of the random search still is not understood correctly. While I agree that it chooses worse models than MetaD2a on average, comparing average to average is a somewhat low bar for a random baseline. However, it seems clear to me that the model zoo is already quite good if choosing just 5 can beat MetaD2a's average score. Possibly it can be considered that the building the model zoo itself is somewhat like a multitask random search initialized on OFA pretraining. That that in itself can obtain fairly good task models from random selection seems important to unwrap.
> > > >
> > > > To be clear that in itself is a strength of your method, but I'm not sure it is sufficiently analyzed and as discussed the comparative training cost should be considered. Your additional experiments have clearly ablated the significant additional significance of the Task conditioned selection algorithm and compared training the competitor methods on the same datasets in a similar fashion. I would currently recommend acceptance of this paper.

---

> > > > > ### Author Response · Authors · 2021-08-29
> > > > > **The random retrieval baseline does not perform purely random network search**
> > > > >
> > > > > We sincerely thank you for your recommendation to accept our paper. The followings are our responses to your questions.
> > > > >
> > > > > ---
> > > > > **“I think the performance of the random search still is not understood correctly. While I agree that it chooses worse models than MetaD2a on average, comparing average to average is a somewhat low bar for a random baseline.“**
> > > > >
> > > > > - There seems to have been some confusion. Please note that the random baseline you suggested, which utilizes a random unseen query dataset for network retrieval, is **not a purely random network search baseline**. This is because the pretrained networks retrieved by this baseline will be ones that **perform well on certain image datasets**, as the cross-modal retrieval space is learned to maximize the (predicted) accuracy on the query dataset. Although such a retrieved network may be suboptimal for the true target dataset, it may not completely fail on the target dataset either, since it may have learned architectures and representations that are task-agnostically beneficial for the general image classification task. For example, the random retrieval baseline can favor a network with skip-connections over ones without the skip-connections, or networks trained on more "generic" datasets.
> > > > >
> > > > > - To see the performance of a **purely random network search** baseline, we should randomly select an arbitrary pretrained network from the entire zoo without any query. However, this model is not an entirely random baseline, since the pretrained networks are trained on image datasets and may have learned representations that are generally beneficial for any image classification tasks, although their architectures may be suboptimal.
> > > > >
> > > > > - Thus, a completely **random baseline** should be one that selects a network with both random architectures and parameters.
> > > > >
> > > > > - We provide the experimental results of these two baselines, namely **random search baseline** which randomly selects a pretrained network from the model zoo, and **purely random baseline**, which selects a random architecture and parameters from the search space, in the table below. As shown, these two baselines largely underperform MetaD2A, and the random retrieval baseline you suggested (which retrieves pretrained networks for a random query dataset).
> > > > >
> > > > > | Method             | Query | Pre-trained | Food Dataset | Gemstones  | Traffic Signs | Dog Breeds | Chinese Char. |
> > > > > |--------------------|--------------|--------------|---------------|------------|---------------|------------|---------------|
> > > > > | TANS (Ours)            | O           | O            |**93.86±0.37**    | **93.88±0.42** | **97.01±0.18**    | **97.24±0.41** | **97.54±0.79**    |
> > > > > | MetaD2A            | -            | -            |89.72±1.35    | 92.72±1.28 | 94.37±2.44    | 96.11±1.47 | 96.87±1.25    |
> > > > > | retrieval with a random query  | O            | O            | 89.16±0.88    | 90.09±3.13 | 93.80±1.44    | 95.03±0.59 | 94.89±0.79    |
> > > > > | random search baseline  | X            | O            | 87.51±1.67    | 90.43±3.11 | 92.03±1.33    | 91.87±3.30 | 93.96±0.38    |
> > > > > | purely random baseline  | X            | X            | 49.30±1.49    | 60.78±7.21 | 70.42±10.54   | 25.21±4.57 | 90.34±3.29    |
> > > > >
> > > > > - Finally, both random baselines significantly underperform our TANS. Please note that MetaD2A is a baseline and not our method.
> > > > >
> > > > > ---
> > > > >
> > > > > **”I'm not sure it is sufficiently analyzed and as discussed the comparative training cost should be considered.**
> > > > >
> > > > > - We will emphasize the difference in the pretraining cost between the baselines and TANS in the revised version of the paper: As shown in Table E.3 in Section E.4 of the supplementary document, the total cost of constructing the 14,000 random pair model zoo is 192 GPU hours. However, unlike with NAS baselines such as PC-DARTS, and DrNAS, **the model zoo construction cost for TANS is a one-time cost**, and once built, TANS can repeatedly reuse the model zoo for **an unlimited number of tasks (datasets)** at **zero extra training cost per each task (dataset)**, while all other baselines require large search cost for every net dataset. Further, we have shown that we can obtain similar levels of performance with smaller model zoos, using the efficient construction method that only adds in near Pareto-optimal models into the model zoo (Table E.3 of the supplementary document).
> > > > >
> > > > > ---
> > > > >
> > > > > We would also like to gently remind you to revise the current score (4) to reflect your positive opinion on our paper, as you seem to have forgotten about it. We thank you again for your helpful feedback and discussions, which we believe have helped significantly strengthen our paper, as well as your time and efforts in reviewing our paper.

---

> > > > > ### Comment · Area_Chair_MgvY · 2021-08-31
> > > > > **Please finalize your score**
> > > > >
> > > > > Dear Reviewer 6K2a,
> > > > > Please finalize your score if you do indeed intend to change it.
> > > > > Thanks,
> > > > > The AC

---

> ### Author Response · Authors · 2021-08-08
> **Response to reviewer 6K2a - Part 1**
>
> **(1) requiring a stronger baseline to demonstrate the generalization of the dataset embedding (for each unseen test dataset X, sample 5-10 random other test datasets, and retrieve and train the optimal models for those datasets on dataset X), making sure if the models selected, but not trained on the specified data still performed adequately on the specified dataset.**
>
> - First, please note that we did not only show the retrieval performance of our TANS framework on the meta-training set, but also have shown that it retrieves models that work well on the given **unseen dataset**in **Figure 6(c)**. Here, we can observe that the closest model to the unseen query dataset has much higher performance on it, compared to the performance of the farthest model. **Figure 4** further shows that despite that there is no instance- class- or dataset-level overlap across the meta-training and meta-test set, TANS retrieves models that are trained on semantically relevant datasets. Finally, **MetaD2A** we used in our main experiment to show the effectiveness of TANS in finding models that are trained on task-relevant datasets for unseen datasets, that is similar to the baselines you mentioned, since it generates *dataset-adaptive* architectures that are trained on a less relevant dataset (ImageNet). We believe that these three sets of results sufficiently show the effectiveness of the generalization power of the proposed cross-modal embedding space, on unseen datasets.
>
> - However, following your suggestion, we additionally conducted the experiments with the baseline you suggested, by using other random **unseen datasets** to retrieve the models. As shown in the table below, TANS consistently outperforms this baseline  (**upto approximately 10%p higher**) with five random unseen datasets used as the queries. These results show that the good performance of TANS is indeed due to its ability to retrieve models trained on relevant datasets, for the **unseen query datasets**. We believe that this baseline will further strengthen the paper and thank you for the helpful suggestion.
>
> |                                   50-Epoch Accuracy on Unseen Target Datasets         ||||||
> |:------------------------------------:|:-------:|:---------:|:-------------:|------------|---------------|
> | **Query**                                |   **Food**  | **Gemstones** | **Traffic Signs** | **Dog Breeds** | **Chinese Char.** |
> | Retrieved by Unseen Random Dataset 1 | 89.82 % |    91.6 % |       94.37 % |    94.42 % |       95.51 % |
> | Retrieved by Unseen Random Dataset 2 | 89.52 % |   93.28 % |       92.96 % |    95.43 % |        95.7 % |
> | Retrieved by Unseen Random Dataset 3 | 87.72 % |   84.87 % |       95.77 % |    95.94 % |       93.47 % |
> | Retrieved by Unseen Random Dataset 4 | 88.62 % |   88.24 % |       94.37 % |    94.42 % |       94.69 % |
> | Retrieved by Unseen Random Dataset 5 | 90.12 % |   92.48 % |       91.55 % |    94.92 % |        95.1 % |
> | **TANS (Ours)**                                 | **94.01 %** |  **94.12 %** |       **97.18 %** |   **96.45 %** |       **96.73 %** |
>
>
> ***
> **(2) None of the existing models were pre-trained on the same training Kaggle datasets and the model zoo was trained on both those and the ImageNet dataset through the OFA initialization**
>
> - Please note that there is no dataset-, class-, and instance-level overlap between the meta-training and the meta-test set (Line 260-261), and the OFA net is trained on an incomparably larger dataset (ImageNet) than any of the models in the model zoo are trained on, since each model is trained only on **a specific Kaggle dataset that is much smaller than ImageNet**. Thus we believe that the baselines are given more advantages in the number of training samples used for training.
>
> - Further, being able to search for relevant pretrained models for a given dataset and utilizing them, is the **unique ability of our method**, and **none**of the existing methods can achieve the same goal. Thus this is an important contribution of our work, rather than an unfair advantage.
>
> - However, to further address your concern regarding comparability of the results, we experimented with the baseline models trained with the same meta-training datasets. We conduct the same meta-test experiments (50 epoch-accuracy) except that the baseline models are **pre-trained on one of the Kaggle datasets used for constructing our model-zoo**, that is randomly selected. For the case of MetaD2A, we meta-trained it on the **exactly the same meta-training set**we used for training TANS. The results in Table A below show that TANS still largely outperforms all baseline models, including MetaD2A, by being able to select the relevant models for a given dataset. We report the 3-run average values for the training times and the test accuracy, along with the 95% confidence intervals.
>
> |    Method   | Params | MFLOPs | Search Time(sec) | Training Times(sec) | Speed Up | Acc. (%) |
> |:-----------:|:------:|:------:|-------------|----------------|----------|----------|
> | MobileNetV3 |   4.21     |  132.94       |   -          |   402.48 ± 99.54           |    1.00x      |    90.03 ± 04.01       |
> | OFA         |      6.00  |     **148.76**   |       121.90      |          500.35 ± 67.07      |     0.65x     |     93.12 ± 01.58     |
> | MetaD2A     |    6.26    |  512.67      |      2.59       |            1228.82 ± 800.38    |     0.32x     |     93.42 ± 01.50     |
> | TANS (Ours) |   **5.51** | 181.74 |      **0.002**  |       **200.93 ± 24.21**  |     **1.98x** |   **96.28 ± 00.24** |
>
> ***
>
> **(3) transfer to common unseen datasets like CIFAR-10/100/ImageNet.**
>
> - The common datasets, such as CIFAR-10, CIFAR-100, and ImageNet, are often well-refined, balanced, and no-biased as they are used for benchmarks in lab settings, and are very much different from the real-world datasets. Since our goal is to build a neural network search methods that works in real-world scenarios, where the given datasets are noisy, have fewer training examples, and possibly biased, we used realistic datasets from Kaggle to simulate such a realistic scenario. Nonetheless, as suggested, we additionally conducted experiments on CIFAR-10 and CIFAR-100, reporting the10-epoch performance of the generated or retrieved models in Table B. We observe that our method still consistently outperforms all base NAS models with a smaller number of parameters and training times, as expected. We will include these results in the revision, along with the results of the ImageNet experiments we could not perform due to the time constraints.
>
> |   CIFAR-10  |          |        |              |                |          |       |
> |:-----------:|:--------:|:------:|--------------|----------------|----------|-------|
> |    **Method**   | **# Params** | **MFLOPs** | **Search Times** | **Training Times** | **Speed Up** | **Acc.**  |
> | MobileNetV3 |     4.21 | 132.94 |            - |         1106.9 |    1.00x | 95.39 |
> |         OFA |     6.00 | **148.76** |      121.90  |       1717.36 |    0.64x |  95.6 |
> |     MetaD2A |     6.26 | 512.67 |         2.59 |        2606.07 |    0.42x | 96.88 |
> | TANS (Ours) |     **5.12** | 200.13 |       **0.0018** |        **1370.01** |    **0.81x** |  **97.1** |
>
> |  CIFAR-100  |          |        |              |                |          |       |
> |:-----------:|:--------:|:------:|--------------|----------------|----------|-------|
> |    **Method**   | **# Params** | **MFLOPs** | **Search Times** | **Training Times** | **Speed Up** | **Acc.**  |
> | MobileNetV3 |    4.321 | 133.16 |            - |         1103.4 |    1.00x | 78.77 |
> |         OFA |     6.00 | **148.76** |      121.90  |       3478.92 |    0.32x |  81.3 |
> |     MetaD2A |     6.26 | 512.67 |         2.59 |        2860.43 |    0.39x | 83.15 |
> | TANS (Ours) |     **5.12** | 200.13 |       **0.0028** |        **1742.00** |   **0.63x** | **84.21** |
>
>
> ***
>
> **(4) compare the models if the flops of each model were included in Table 1.**
>
> - Thank you for the helpful suggestion. In Table C below, we report the FLOPs of all methods from Table 1. We will include this column to Table 1 in the revision.
>
> | Method | MobileNetV3 |  DrNAS | PC-DARTS | FB-Net | OFA    | MetaD2A | TANS (Ours) |
> |:------:|:-----------:|:------:|----------|--------|--------|---------|-------------|
> | MFLOPs |     132.94  | 623.43 |   566.55 | 246.69 | 148.76 |  512.67 | 181.74      |
>
>
> ***
>
> **(5) how much compute is used to generate the Model zoo.**
>
> - We did report the computational cost of the model zoo construction in Section E.4 (Table E.3) of the supplementary file. The total cost of constructing the model zoo with 14,000 pretrained models is 192 GPU hours. Using the efficient model zoo construction method, however, the model zoo construction will only take as little as 1/14 of the cost of constructing the full model zoo with 14K pretrained models, as shown in Figure 6 (d). We will include this in the main paper, if space allows.

---

### Official Review · Reviewer_orau · 2021-07-18

**Rating:** 7
**Confidence:** 4

**Summary:**

This paper proposes a new Task-Adaptive Neural network search framework (TANS). Such a task needs to retrieve a good network in well pre-trained dataset-network sets. In order to obtain a task-adaptive neural network, the amortized meta-learning method is employed. Meanwhile, this meta-learning framework is trained with contrastive learning to maximize the similarity between the positive dataset-network pairs and minimize the similarity between the negative pairs. The experiments show that TANS can rapidly retrieve a well-fitted network for unseen datasets.

**Limitations And Societal Impact:**

There are still some unclear descriptions in this manuscript. The specific contents are shown in the Main review. Judging from the paper, there is currently no direct negative impact.

**Main Review:**

Pros:
(1)  The authors propose a new task, which retrieves a good network in well pre-trained dataset-network sets. Such a task is promising and novel.
(2)  A meta-learning framework is trained with contrastive learning to maximize the similarity between the positive dataset-network pairs and minimize the similarity between the negative pairs.
(3)  The experiments in ten real-world datasets show the effectiveness of TANS.

Cons:
I understand the novel task of this manuscript. I have some questions as follows. I want to make sure again, is it the first work to propose architecture retrieval? I think this task is very promising because we can hardly search for a suitable network given any task. The second question is if the dataset is very large, e.g., ImageNet, how to sample images in this dataset? What kind of image is the suitable image that represents this dataset? Whether the simple random sampling method is applicable? Another question is whether such a method only tackles the image classification task. What if object detection or other regression tasks?

To sum up, this manuscript introduces a new task, which is valuable in real-world architecture retrieval. The writing is clear and easy to understand. However, there are still some unclear descriptions and questions. Some tables and figures (e.g., Figure 7, Table 3) are also too small to see results clearly.


**Time Spent Reviewing:**

4 hours

---

> ### Author Response · Authors · 2021-08-09
> **Response to Reviewer orau**
>
> **(1) I want to make sure again, is it the first work to propose architecture retrieval? I think this task is very promising because we can hardly search for a suitable network given any task.**
>
> - We sincerely appreciate your comment that our work is very promising. To the best of our knowledge, there does not exist any works that aim to retrieve the best-performing pretrained networks for a given query dataset, and we believe that this is due to the difficulty of encoding datasets and networks. The latter is especially challenging due to the needs of encoding the parameters of a network, which is not straightforward, and we tackle this by utilizing both the topology and functional embedding.
>
> ***
>
> **(2) The second question is if the dataset is very large, e.g., ImageNet, how to sample images in this dataset? What kind of image is the suitable image that represents this dataset? Whether the simple random sampling method is applicable?**
>
> - This is a very insightful question. When the dataset is very large, we can use the hierarchical set encoder proposed in MetaD2A that first encodes each class by set-encoding its element, and then further encodes the set of class embeddings into a dataset embedding. This enables to effectively encode large-scale datasets. If the dataset is even larger such that all instances from each class does not fit into the memory, we can further split the set of instances for the class into subsets, encode each subset, and then set-encode these subset embeddings.
> - We also empirically found that simple random sampling works to a certain extent.
> - Finally, in practical scenarios, most users would not want to upload their entire datasets, due to privacy concerns.
>
> ***
>
>
> **(3) Another question is whether such a method only tackles the image classification task. What if object detection or other regression tasks?**
>
> - Our framework is compatible with any neural networks and tasks, regardless of the domains, as long as we can properly encode the architectures and the datasets, and measure their accuracies. Thus we expect TANS to work with highly different domains, such as speech recognition, although building a dataset encoder for speech may require nontrivial efforts. Since most object detectors and neural networks for regression can be also embedded with our topology and functional embedding, and the datasets can be encoded in the same manner, we believe that TANS can be applied to such tasks without any problem. We will perform and include the results of our framework on object detection or semantic segmentation tasks in the final revision, to further demonstrate the general applicability of our framework on various types of tasks. However, please understand that it is not possible to perform this experiment with one week's time given for the rebuttal period, as it would require designing the experiments and constructing the model zoo from scratch.
>
> ***
>
> **(4) Some tables and figures (e.g., Figure 7, Table 3) are also too small to see results clearly.**
>
> - Thank you for your helpful suggestions for improving the legibility of the tables and figures. We will adjust the resolution and the ratio of Figure 7, and the size of Table 3, for the revision.

---

### Official Review · Reviewer_5Ssa · 2021-07-20

**Rating:** 8
**Confidence:** 4

**Summary:**

The paper is about retrieving a pre-trained network from a model zoo in a computationally effective manner. The authors proposed a network topology and parameter embedding that can be used to retrieve networks that perform well on a task represented by a dataset after fine-tuning.

**Ethical Concerns:**

No concerns

**Ethics Review Area:**

["I don’t know"]

**Limitations And Societal Impact:**

Yes

**Main Review:**

I find the paper to be clearly written, interesting to read, the topic to be relevant and the contribution to be novel and significant. The paper contains a significant methodological contribution, defining a new task consisting of multiple datasets. The code is provided for reproducibility. A quick inspection shows that the code is well organised and contains all the ingredients to download and prepare the data.  I recommend accept and provide minor comments below.

- line 125, it took me some time to understand why neural network is denoted $M^\tau$. I realized later on that $M$ refers to model. Indeed, the proposed method could be generalized to arbitrary models, why not make it clear in line 125?
- it is unclear to me why authors chose a title that is different from the name of their core contribution. Why not have a title "TANS: Task-Adaptive Network Search." or "TANS: Task-Adaptive Network Search with Meta-Contrastive Learning." ?
- what is the total compute capacity that is needed to reproduce the results?
- datasets used in the study are public, but it would be a good idea to provide them from one location, because they may disappear with time. Have you thought about it?

**Time Spent Reviewing:**

1 hour

---

> ### Author Response · Authors · 2021-08-09
> **Response to Reviewer 5Ssa**
>
> **(1) line 125, it took me some time to understand why neural network is denoted M^T. I realized later on that M refers to model. Indeed, the proposed method could be generalized to arbitrary models, why not make it clear in line 125?**
>
> - Thank you for your thoughtful suggestion. We will clarify and specify that the notation $M$ denotes arbitrary models (Line 125) in our final revision.
>
> ***
>
> **(2) It is unclear to me why authors chose a title that is different from the name of their core contribution. Why not have a title "TANS: Task-Adaptive Network Search." or "TANS: Task-Adaptive Network Search with Meta-Contrastive Learning." ?**
>
> - We thank you for kindly suggesting a better title for our paper. Following your suggestion, we will revise the title of the paper to ‘"TANS: Task-Adaptive Neural Network **Search**with Meta-Contrastive Learning".
>
> ***
>
> **(3) What is the total compute capacity that is needed to reproduce the results?**
>
> - As shown in Table E.3 in Section E.4 of the supplementary document, the total cost of constructing the 14,000 random pair model zoo is 192 GPU hours. However, this is a one-time cost, and we can obtain similar levels of performance with smaller model zoos, when using the efficient model zoo construction method (please see Table E.3 of the supplementary document).
>
> ***
>
> **(4) Datasets used in the study are public, but it would be a good idea to provide them from one location, because they may disappear with time. Have you thought about it?**
>
> - We will open-source our model-zoo (including datasets and pre-trained models) and the code for the retrieval framework such that anyone interested in our framework can freely access them. Further, we plan to build an online platform on which anybody can contribute their pretrained models and download the most relevant models for a given dataset and constraints (e.g. computational and memory cost). Please see Section F.1 of the supplementary document for more discussions on this topic.

---

> > ### Comment · Reviewer_5Ssa · 2021-08-26
> > **My score stands**
> >
> > I thank the authors for their comprehensive responses. I read all reviewer comments and all author responses. I believe that the authors provided a very effective rebuttal response. My score stands and I wish the authors the best of luck.

---

### Author Response · Authors · 2021-08-10
**Summary of the responses from all reviewers**

We sincerely appreciate your time and effort in reviewing our paper, as well as the positive feedback that our work is **novel and promising** from all reviewers. During the rebuttal period, we have faithfully addressed all the concerns from the reviewers, conducting multiple sets of experiments to respond to the comments from reviewer **6K2a** and **ZAGc**, as follows:

1. **6K2a**: stronger baseline to demonstrate the generalization of the dataset embeddings
2. **6K2a**: comparison with baselines pretrained on Kaggle datasets
3. **6K2a**: transferring to standard benchmark datasets (CIFAR-10/100)
4. **ZAGc**: additional ablation study for functional embedding
5. **ZAGc**: additional ablation study for performance predictor
6. **ZAGc**: correlation between latent distance and performance

We thank you for your helpful suggestions, as we found the new discussions and experimental results to be highly valuable, which further strengthens our paper. We want to finally emphasize again that exploring the unique and novel challenges of **Neural Network Search (NNS)** and the proposal of novel meta-learning framework to tackle the problem (**Task-Adaptive Neural Network Search with Meta-Contrastive Learning**) are both highly novel contributions with potentially large practical impact.

---

### Decision · Program_Chairs · 2021-09-27

**Decision:**

Accept (Spotlight)

**Comment:**

A strong paper on a subject that many people in the community are likely to find interesting.  Multiple reviewers agreed that this submission could be a Top-50 paper. But it was felt that the work could provide a better analysis, and possibly further experiments on the model zoo creation and corresponding properties. A healthy discussion occurred during the author response and discussion phase. New experiments and further explanations from the authors motivated one of the reviewers to upgrade their score and assessment of the paper to an accept, leading to all reviewers recommending acceptance. More discussions of comparative pretraining costs and additional experiments provided by the authors during the rebuttal helped to demonstrate the robustness of the method on other datasets.

The work addresses a subject that the reviewers felt is interesting and important, proposing a network topology and parameter embedding that can be used to retrieve networks that perform well on a task represented by a dataset after fine-tuning. The paper proposes an amortized meta-learning framework to learn a cross-modal latent space using a contrastive loss. Their goal is to maximize the similarity between a dataset and a high-performing network on it, and minimize the similarity between irrelevant dataset-network pairs.

Given the good reception by the reviewers after the author response and the good level of discussion about this work, the AC recommends a Spotlight.